# Spatially compartmentalized phase regulation of a Ca²⁺-cAMP-PKA oscillatory circuit

Brian Tenner[1,2], Michael Getz[3], Brian Ross[2], Donya Ohadi[4], Christopher H Bohrer[1], Eric Greenwald[2], Sohum Mehta[2], Jie Xiao[1], Padmini Rangamani[3,4]*, Jin Zhang[2,5]*

[1]Department of Biophysics and Biophysical Chemistry, The Johns Hopkins University School of Medicine, Baltimore, United States; [2]Department of Pharmacology, University of California, San Diego, La Jolla, United States; [3]Chemical Engineering Graduate Program, University of California, San Diego, La Jolla, United States; [4]Department of Mechanical and Aerospace Engineering, University of California, San Diego, La Jolla, United States; [5]Department of Chemistry and Biochemistry, University of California, San Diego, La Jolla, United States

**Abstract** Signaling networks are spatiotemporally organized to sense diverse inputs, process information, and carry out specific cellular tasks. In β cells, Ca²⁺, cyclic adenosine monophosphate (cAMP), and Protein Kinase A (PKA) exist in an oscillatory circuit characterized by a high degree of feedback. Here, we describe a mode of regulation within this circuit involving a spatial dependence of the relative phase between cAMP, PKA, and Ca²⁺. We show that in mouse MIN6 β cells, nanodomain clustering of Ca²⁺-sensitive adenylyl cyclases (ACs) drives oscillations of local cAMP levels to be precisely in-phase with Ca²⁺ oscillations, whereas Ca²⁺-sensitive phosphodiesterases maintain out-of-phase oscillations outside of the nanodomain. Disruption of this precise phase relationship perturbs Ca²⁺ oscillations, suggesting the relative phase within an oscillatory circuit can encode specific functional information. This work unveils a novel mechanism of cAMP compartmentation utilized for localized tuning of an oscillatory circuit and has broad implications for the spatiotemporal regulation of signaling networks.

*For correspondence:
padmini.rangamani@eng.ucsd.edu (PR);
jzhang32@ucsd.edu (JZ)

## Introduction

Cyclic adenosine monophosphate (cAMP) and Ca²⁺ act as essential second messengers in almost every cell type and regulate many functional pathways within a cell, such as hormonal signal transduction, metabolism, and secretion (*Clapham, 2007*; *Sassone-Corsi, 2012*). In some cell types, including neurons, cardiomyocytes, and pancreatic β cells, these messengers' concentrations oscillate intracellularly (*Dupont et al., 2011*; *Dyachok et al., 2006*), and the oscillations encode critical signaling information (e.g. signal strength, duration, and target diversity) into parameters such as frequency and amplitude (*Berridge et al., 1998*; *De Pittà et al., 2008*; *Parekh, 2011*). This phenomenon is perhaps best exemplified in β cells, where oscillations of Ca²⁺ drive pulsatile insulin secretion (*Rorsman and Ashcroft, 2018*) and oscillating cAMP levels (*Nesher et al., 2002*; *Tengholm, 2012*). Furthermore, Ca²⁺, cAMP, and the downstream cAMP-dependent kinase protein kinase (PKA) constitute a highly coordinated oscillatory circuit responsible for integrating metabolic and signaling information (*Ni et al., 2011*). In addition to temporal control, biochemical pathways are also spatially organized within the cell (*Smith and Scott, 2002*; *White and Anderson, 2005*). Both Ca²⁺ and cAMP are highly spatially compartmentalized and form signaling microdomains or nanodomains (*Calebiro and Maiellaro, 2014*; *Petersen, 2002*). While Ca²⁺ levels are locally controlled by

channels, pumps, and intracellular buffering systems (*Clapham, 2007*; *Stern, 1992*), cAMP is thought to be regulated via controlled synthesis by adenylyl cyclases (ACs) and degradation by phosphodiesterases (PDEs) (*Bender and Beavo, 2006*; *Defer et al., 2000*). Despite extensive studies on cAMP compartmentation, the mechanisms that spatially constrain this mobile second messenger await to be fully elucidated (*Saucerman et al., 2014*; *Lohse et al., 2017*; *Musheshe et al., 2018*; *Bock et al., 2020*; *Zhang et al., 2020*). Furthermore, it is not clear how the spatial regulation of a second messenger influences its dynamic behaviors in the context of coordinated oscillations.

In this study, we investigated the spatiotemporal organization of the $Ca^{2+}$-cAMP-PKA oscillatory circuit in MIN6 β cells and discovered that the relative oscillatory phase between cAMP/PKA and $Ca^{2+}$ is spatially regulated within signaling nanodomains. By combining dynamic live-cell imaging, super-resolution microscopy, and computational modeling, we further found that fine-scale, compartment-specific perturbations of this precise phase regulation impact $Ca^{2+}$ oscillations in β cells. These findings suggest that the relative phase in oscillatory signaling circuits, like the amplitude and frequency, represents yet another mode of informational encoding and processing, which is subjected to spatiotemporal regulation within the cell.

## Results

### The relative phase of β cell cAMP and $Ca^{2+}$ oscillations is compartmentalized

In order to study the spatiotemporal relationship between key players of the $Ca^{2+}$-cAMP-PKA circuit, we chose to focus our attention on an important class of molecular scaffolds, A-kinase anchoring proteins (AKAPs), which are responsible for recruiting PKA to specific substrates at distinct subcellular locations. In several excitable cell types, the plasma membrane (PM)-localized scaffold protein AKAP79 (rodent ortholog AKAP150) has been shown to organize a macromolecular complex with binding partners that include PKA, the voltage-gated $Ca^{2+}$ channel $Ca_V1.2$, Protein Kinase C (PKC), the $Ca^{2+}$/calmodulin-dependent protein phosphatase calcineurin, $Ca^{2+}$-sensitive ACs, AMPA receptors, and many others (*Gold et al., 2011*). Due to the extensive and multivalent nature of AKAP79/150 (as the scaffold is commonly referred) and a report describing the functional impairment of glucose-stimulated insulin secretion (GSIS) in pancreatic β cells upon its knock-out (*Hinke et al., 2012*), we hypothesized that the AKAP79/150 scaffold might play an important role in the spatiotemporal regulation of the $Ca^{2+}$-cAMP-PKA oscillatory circuit. Specifically, we were interested in testing if AKAP79/150 is able to create a spatially-distinct compartment in which recruitment of signaling effectors can locally fine-tune and reshape signaling dynamics within the circuit (*Beene and Scott, 2007*; *Greenwald and Saucerman, 2011*).

To test this hypothesis, we monitored intracellular cAMP and $Ca^{2+}$ using the FRET-based cAMP biosensor (Ci/Ce)Epac2-camps (*Everett and Cooper, 2013*) and the red $Ca^{2+}$ indicator RCaMP (*Akerboom et al., 2013*). We measured cAMP concentration changes in the immediate vicinity of AKAP150 in mouse MIN6 β cells by using (Ci/Ce)Epac2-camps fused to the full-length AKAP79 scaffold (gene *AKAP5*) (*Figure 1a*). Although human AKAP79 and the rodent AKAP150 share only 53% sequence identity overall, the interaction motifs and association between the AKAP scaffold and other key signaling players such as PKA, the voltage-gated calcium channel, adenylyl cyclases, and calcineurin are highly conserved (*Willoughby et al., 2010*; *Zhang et al., 2016*). The primary difference between these two closely related scaffolds is the presence of an internal repetitive amino acid sequence of unknown function in AKAP150 (*Robertson et al., 2009*). The functional equivalence of AKAP79 and AKAP150 has been demonstrated in several reciprocal knock-out and recovery experiments in which AKAP79 or AKAP150 was knocked out and expression of the other was shown to rescue a measured phenotype (*Hoshi et al., 2005*; *Zhang et al., 2011*). As a control, we also targeted the cAMP probe to the general plasma membrane by adding a lipid modification domain (*Wachten et al., 2010*). These targeted biosensors allowed us to compare the dynamics within the AKAP79/150-specific compartment versus the general plasma membrane compartment (*Figure 1a*).

Although both targeted sensors were trafficked to and distributed along the plasma membrane (*Figure 1—figure supplement 1a,b*), we observed notable differences in their respective cAMP signals relative to $Ca^{2+}$ oscillations after triggering the circuit (*Figure 1b*) with tetraethylammonium chloride (TEA, 20 mM), a potent $K^+$ channel blocker. cAMP oscillations measured within the

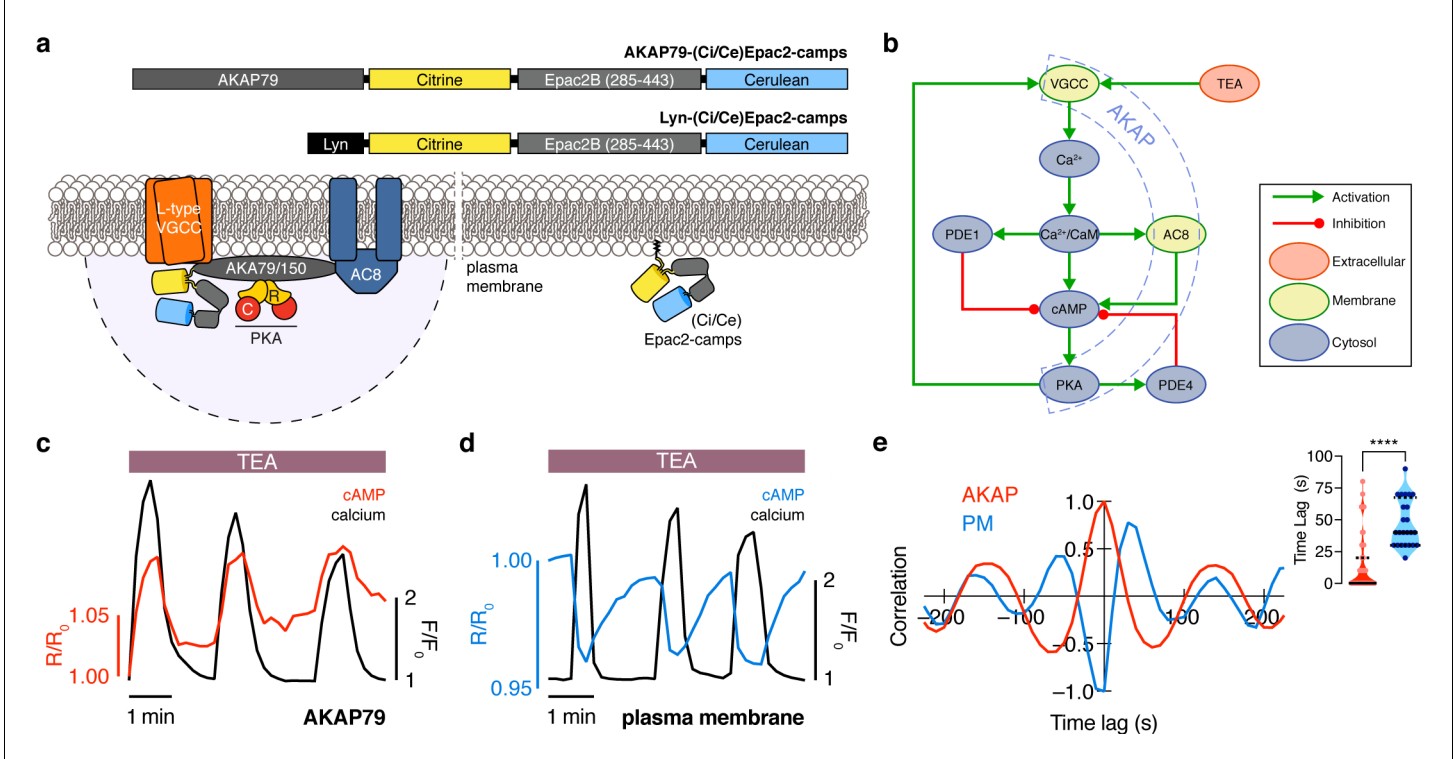

**Figure 1.** The phase of oscillating cAMP is shifted between the AKAP79/150 compartment and the general plasma membrane compartment, relative to $Ca^{2+}$. (a) Depiction of the AKAP79/150 and plasma membrane compartments, including the targeted cAMP biosensor (Ci/Ce)Epac2-camps to measure the compartment-specific cAMP signaling. Schematics of the lyn-(Ci/Ce)Epac2-camps and AKAP79-(Ci/Ce)Epac2-camps sensors are shown above. (b) Network diagram describing the key players in the β cell $Ca^{2+}$-cAMP-PKA oscillatory circuit. (c) Representative single-cell trace of in-phase oscillating responses of AKAP79-(Ci/Ce)Epac2-camps and RCaMP, whole-cell fluorescence measured. Red trace is cAMP (cyan direct channel divided by CY-FRET channel) and black trace is $Ca^{2+}$ (RFP). (d) Representative single-cell trace of out-of-phase oscillating responses of lyn-(Ci/Ce)Epac2-camps and RCaMP, whole-cell fluorescence measured. Blue trace is cAMP (cyan direct channel divided by CY-FRET channel) and black trace is $Ca^{2+}$ (RFP). (e) Cross-correlation between the oscillatory $Ca^{2+}$ and cAMP signals from the representative in-phase AKAP79 (red) and out-of-phase plasma membrane (PM, blue) responses from c, d. Time lag (sec) between the cAMP and $Ca^{2+}$ signals for the two compartments (AKAP79/150, red, is 13 ± 3 sec n=60 and PM, blue, is 47 ± 4 s n=24). ****p<0.0001; unpaired two-tailed Student's t-test.

The online version of this article includes the following figure supplement(s) for figure 1:

**Figure supplement 1.** The AKAP79/150 and PM-targeted sensors are localized at the plasma membrane.
**Figure supplement 2.** Cytosolic and AKAP79-specific cAMP oscillations are anti-correlated in the same cell.

AKAP79/150 compartment were in-phase with oscillating $Ca^{2+}$, such that each transient spike in intracellular $Ca^{2+}$ was closely associated with a transient increase in cAMP (*Figure 1c*) (n = 60 cells). This was in sharp contrast to cAMP oscillations measured within the general plasma membrane compartment, where each local $Ca^{2+}$ peak corresponded to a local trough in cAMP (n = 24), followed by a slow reversal of both signals to a pre-stimulated baseline (*Figure 1d*). While these out-of-phase cAMP-$Ca^{2+}$ oscillations were consistent with those observed in the cytoplasm of β cells (*Landa et al., 2005*; *Ni et al., 2011*), in-phase cAMP-$Ca^{2+}$ oscillations had not previously been observed under these conditions. To quantify the cAMP-$Ca^{2+}$ phase relationship, we measured the lag time by calculating the cross-correlation between the two normalized, oscillatory signals and finding the shortest delay yielding the maximum correlation (see Appendix 1 for details) (*Figure 1e*). In-phase cAMP oscillations corresponded to short lag times (typically <20 s), while out-of-phase oscillations mostly possessed longer lag times. Within the AKAP79/150 compartment, cAMP lagged behind $Ca^{2+}$ by an average of only 13 ± 3 s (mean ± SEM, n = 60); however, cAMP within the general plasma membrane compartment oscillated with a lag time of 47 ± 4 s (n = 24) behind $Ca^{2+}$ (*Figure 1e*). In order to measure these different cAMP dynamics within the same cells, we transfected MIN6 cells with two cAMP sensors of different targeting sequences and colors, AKAP79-(Ci/

Ce)Epac2-camps and the red cytosolic cAMP sensor R-FlincA (*Ohta et al., 2018*). After stimulating the signaling circuit, we observed anti-correlated cAMP oscillations, indicating that cAMP oscillates with different phases between the cytosol/plasma membrane and the immediate vicinity of AKAP79/150 within the same cell (n = 25) (*Figure 1—figure supplement 2*). This stark difference in the cAMP-$Ca^{2+}$ phase relationship suggests that the relative phase of this oscillatory circuit is compartmentalized and hints at differential regulation of the circuit between the AKAP79/150 compartment and the general plasma membrane compartment.

## Oscillatory phase is regulated by balanced activities of $Ca^{2+}$-sensitive ACs and PDEs

Given that in-phase cAMP oscillations were only observed within the AKAP79/150 compartment (*Figure 1c*) and that out-of-phase cAMP oscillations were observed in the general plasma membrane compartment (*Figure 1d*) and the cytoplasm (*Ni et al., 2011*), we hypothesized that $Ca^{2+}$ oscillations are coupled to cAMP oscillations by a ubiquitous mechanism throughout the cell, while additional mechanisms specifically regulate the phase relationship within the AKAP79/150 compartment. We first sought to identify the component that is responsible for coupling cAMP dynamics to $Ca^{2+}$ dynamics globally. Since TEA induces continuous $Ca^{2+}$ oscillations, we determined the temporal relationship between $Ca^{2+}$ and cAMP at the general plasma membrane more precisely by measuring the impulse response of the circuit following a transient membrane depolarization. After the addition of KCl (15 mM) followed by a subsequent washout to elicit a transient influx of $Ca^{2+}$ (*Dou et al., 2015*), we observed a synchronous cAMP decrease (n = 20) followed by a return to baseline (*Figure 2a*). These data suggest that increasing cytosolic $Ca^{2+}$ was coupled to a decrease in cAMP at the plasma membrane through $Ca^{2+}$-sensitive AC or PDE activities. Given that $Ca^{2+}$-inhibited ACs (AC5, AC6) have low specific activity in both the presence and absence of physiological $Ca^{2+}$, as well as relatively low expression in the pancreas (*Defer et al., 2000*), we instead focused on probing the role of PDEs. The $Ca^{2+}$-dependent PDE1 family in β cells, specifically PDE1C, has been implicated in modulating GSIS (*Han et al., 1999*). Indeed, acute addition of 8MM-IBMX (100 μM), a relatively selective PDE1 inhibitor, effectively uncoupled cAMP dynamics from $Ca^{2+}$ oscillations (*Figure 2b*, *Figure 2—figure supplement 1a*) (n = 18), indicating that $Ca^{2+}$-triggered activation of PDE1 mediates the transient cAMP decreases. We also observed that the overall increase in cAMP led to an increase in the $Ca^{2+}$ oscillation frequency, consistent with the previously identified role of cAMP/PKA in regulating the $Ca^{2+}$ oscillations (*Ni et al., 2011*). We tested the roles of two additional families of abundant PDEs in pancreatic β cells, PDE3 and PDE4, by acute pharmacologic inhibition. While treating cells with either milrinone (PDE3 inhibitor, 10 μM, n = 12) or rolipram (PDE4 inhibitor, 1 μM, n = 15) slightly increased cAMP levels, neither inhibitor had an effect on cAMP-$Ca^{2+}$ coupling or relative phase (*Figure 2—figure supplement 1b,c*). These data suggest that PDE1 is the key component that couples $Ca^{2+}$ and cAMP oscillations within this signaling circuit.

How is the phase relationship between $Ca^{2+}$- and cAMP-regulated within distinct signaling compartments? In order to gain a more quantitative understanding of the regulation of the cAMP-$Ca^{2+}$ phase relationship, we created a simplified well-mixed mathematical model involving $Ca^{2+}$, cAMP, and $Ca^{2+}$-driven PDE and AC activity components (*Cooper et al., 1995*; *Figure 2c*, see Appendix 1 for details). This simple circuit represents the key aspects of the oscillatory cAMP-$Ca^{2+}$ circuit and is applicable to different signaling compartments. Opposite to the $Ca^{2+}$-stimulated PDE1 (*Ang and Antoni, 2002*) is the $Ca^{2+}$-stimulated AC8 (gene *Adcy8*) (*Masada et al., 2009*; *Masada et al., 2012*), an abundant $Ca^{2+}$-sensitive transmembrane AC isoform in β cells that has been shown to mediate sustained insulin secretion and associate with the AKAP79/150 scaffold (*Dou et al., 2015*; *Willoughby and Cooper, 2006*; *Willoughby et al., 2010*). By computationally manipulating the activity of each arm, we found that cAMP can oscillate either out-of-phase or in-phase when a $Ca^{2+}$ pulse train is used as an input (*Figure 2c*). In particular, when the relative activity of PDE1 is greater than the activity of AC8, $Ca^{2+}$-driven cAMP degradation dominates, resulting in an out-of-phase cAMP-$Ca^{2+}$ relationship. On the other hand, if the relative activity of AC8 is greater than that of PDE1, $Ca^{2+}$-stimulated cAMP production is favored, and an in-phase relationship is observed, consistent with previous modeling studies (*Fridlyand et al., 2007*; *Peercy et al., 2015*).

Thus, our simplified model indicates that the phase relationship can be tuned by altering the relative strength between $Ca^{2+}$-sensitive ACs and PDEs (*Figure 2c*). This model provided a blueprint for understanding the interplay between the $Ca^{2+}$-stimulated AC/PDE balance and the cAMP-$Ca^{2+}$

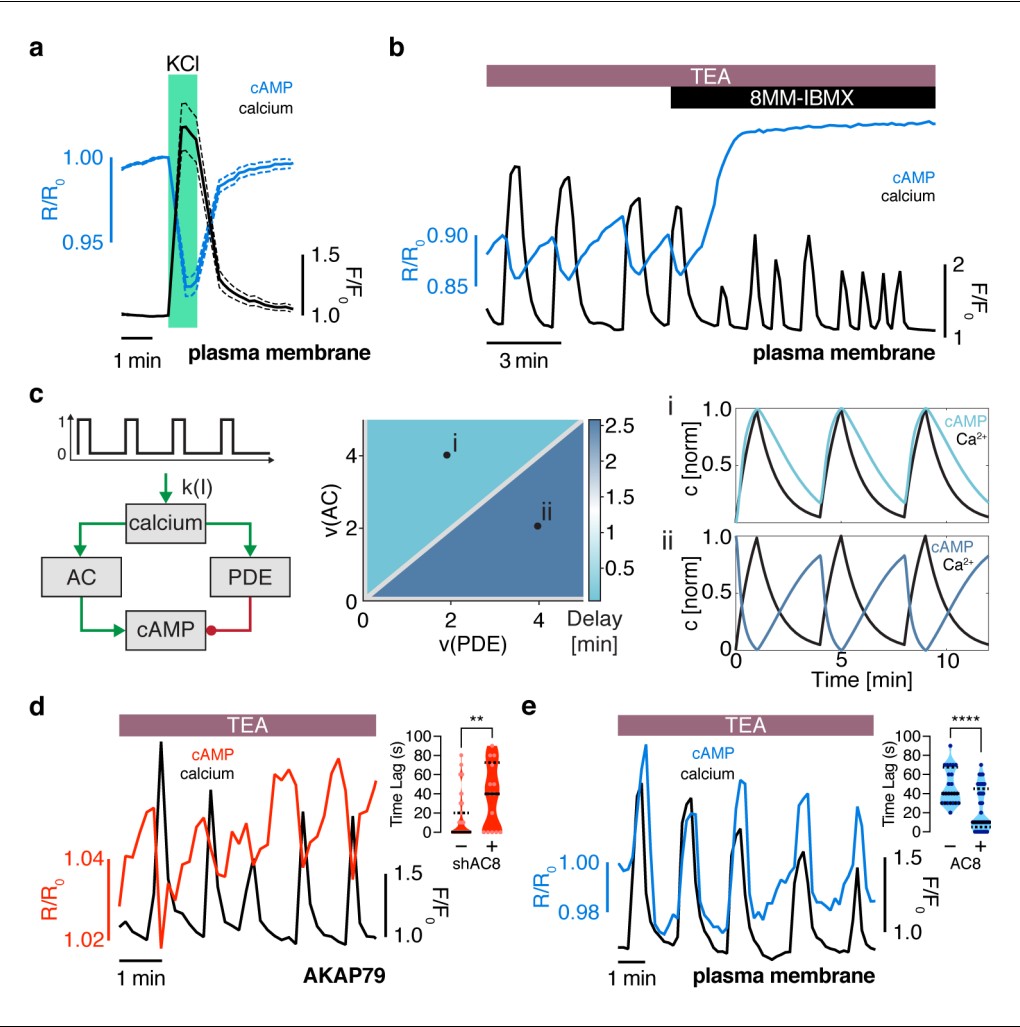

**Figure 2.** The oscillation phase is regulated by a balance between Ca²⁺-sensitive AC and PDE activity. (**a**) Impulse response of plasma membrane cAMP (blue) to a spike in Ca²⁺ entry (black), triggered by KCl-mediated membrane depolarization (wash in/out). The transient decrease in PM-cAMP is coupled to the transient increase in intracellular Ca²⁺. (**b**) Acute inhibition of Ca²⁺-sensitive PDE1 decouples the out-of-phase PM-cAMP oscillations from Ca²⁺ oscillations, as observed in this representative cell trace (Ca²⁺ – black, PM-cAMP – blue). (**c**) The oscillatory phase of cAMP can be manipulated by tuning the relative activity of Ca²⁺-sensitive PDE and AC, as demonstrated by a simplified mathematical model. The schematic shows the network architecture. The relative activities of AC and PDE, denoted as v(AC) and v(PDE), control the delay between Ca²⁺ and cAMP. Point (**i**), with high AC activity and low PDE, shows in phase oscillations of Ca²⁺ and cAMP, whereas point (**ii**), with high PDE and low AC activity shows out of phase oscillations. These dynamics are shown in the line graphs. (**d**) Knocking down AC8 is correlated with an increase in the time lag for oscillatory cAMP at the AKAP79/150 microdomain (37 ± 9 s, n = 11), indicating more cells exhibiting out-of-phase cAMP oscillations (representative cell trace, Ca²⁺ – black, AKAP79/150-cAMP – red). (**e**) Over-expressing AC8 is sufficient to reverse the phase at the PM to in-phase (23 ± 2 s, n = 56) (representative cell trace, Ca²⁺ – black, PM cAMP – blue). **p=0.0014, ****p<0.0001; unpaired two-tailed Student's t-test.

The online version of this article includes the following figure supplement(s) for figure 2:

**Figure supplement 1.** The Ca²⁺-dependent cAMP response is dependent on PDE1, but not PDE3 or PDE4.
**Figure supplement 2.** The cAMP-Ca²⁺ phase relationship at the PM can be tuned by expression of the Ca²⁺-dependent AC8.
**Figure supplement 3.** No difference in endogenous AC8 levels is observed between cells expressing AKAP79-(Ci/Ce)Epac2-camps and lyn-(Ci/Ce)Epac2-camps.

phase relationship within the AKAP79/150 compartment. Based on the findings from our model, we predicted that decreasing the relative contribution of AC8 will shift the cAMP-Ca$^{2+}$ phase relationship from in-phase to out-of-phase. To test this prediction, we knocked down endogenous AC8 in MIN6 cells as previously done (*Raoux et al., 2015*) and observed that most cells exhibited out-of-phase cAMP oscillations within the AKAP79/150 compartment (average lag time 37 ± 9 s, n = 11) (*Figure 2d*), indicating an AC8-specific role in mediating the cAMP-Ca$^{2+}$ phase signature.

Conversely, increasing the relative contribution of AC8, for example, by increasing the concentration of AC8 throughout the PM, should shift the cAMP-Ca$^{2+}$ phase relationship from out-of-phase to in-phase. To test this prediction, we overexpressed full-length AC8 and examined the effect in the general plasma membrane compartment. Interestingly, we found that AC8 overexpression reversed the out-of-phase cAMP-Ca$^{2+}$ phase relationship in a titratable manner, where the percentage of in-phase oscillating cells correlated with increasing amounts of co-transfected AC8 (average lag time 23 ± 2 s, n = 56) (*Figure 2e*, *Figure 2—figure supplement 2a–d*). In order to examine the effects of the targeted cAMP biosensors on AC8 expression, we also measured endogenous AC8 levels in cells transfected with either lyn-(Ci/Ce)Epac2-camps or AKAP79-(Ci/Ce)Epac2-camps and observed no significant difference (*Figure 2—figure supplement 3a,b*). These data demonstrate that higher levels of AC8 are sufficient to reverse the cAMP phase at the plasma membrane. In summary, these phase-manipulation experiments suggest that the cAMP-Ca$^{2+}$ phase relationship is representative of a sensitive, compartmentalized balance between the Ca$^{2+}$-stimulated activities of PDE1 and AC8.

## Membrane-localized AKAP150:AC8 nanoclusters regulate cAMP-Ca$^{2+}$ oscillatory phase

The close spatial juxtaposition between the AKAP79/150 and general plasma membrane compartments presents a significant challenge for cAMP compartmentation, in that cAMP oscillations must be distinctly regulated within these adjacent signaling domains. Indeed, how cAMP, a rapidly diffusing small molecule, is spatially compartmentalized in cells is not yet completely understood, especially given the relatively low catalytic efficiency of a single cAMP-producing AC and degrading PDE (*Conti et al., 2014*; *Lohse et al., 2017*; *Bock et al., 2020*; *Zhang et al., 2020*). Given that AKAP79/150 exists in nanoclusters at the plasma membrane in multiple cell types (*Mo et al., 2017*; *Zhang et al., 2016*) and associates with AC8 in β cells (*Willoughby et al., 2010*), we hypothesized that AC8 could form nanoclusters on the plasma membrane of MIN6 cells and compartmentalize cAMP dynamics. To test this hypothesis, we examined the spatial organization of AC8 and AKAP150 at the membrane using Stochastic Optical Reconstruction Microscopy (STORM). We found that AKAP150 molecules were organized in clusters with a mean radius of 127 ± 9 nm and an average nearest-neighbor spacing of 313 ± 20 nm between cluster centers (n = 20) (*Figure 3a* and *Figure 3—figure supplement 1*), consistent with several recent reports demonstrating AKAP79/150's tendency to form nanoclusters in other cell types (*Mo et al., 2017*; *Purkey et al., 2018*; *Tajada et al., 2017*; *Zhang et al., 2016*). Thus, the AKAP79/150 compartment-specific cAMP phase is likely representative of the balanced cAMP generation and degradation within these AKAP clusters.

Due to the known interaction between AKAP79/150 and AC8 (*Willoughby et al., 2010*) and diffusion of membrane-localized signaling complexes, we next probed the spatial organization and mobility of AC8. We found AC8 also distributes non-uniformly at the plasma membrane and forms clusters with a mean radius of 88 ± 8 nm and an average nearest-neighbor spacing of 292 ± 16 nm between cluster centers (n = 16) (*Figure 3b*; *Figure 3—figure supplement 1*). In order to measure the degree of colocalization between AKAP150 and AC8 nanoclusters, we also performed 2-color STORM and observed 72% of AKAP150 localizations were co-clustered with AC8 localizations, suggesting the presence of AKAP150:AC8 co-clustered nanodomains (*Figure 3—figure supplement 2a–e*). Next, to determine the mobility of AC8 molecules within the timescale of the oscillatory circuit's period, we performed Fluorescence Recovery After Photobleaching (FRAP) with EGFP-tagged AC8 and found AC8 diffuses slowly at the membrane, and that there exists a significant immobile fraction ($D_{AC8}$ = 0.019 ± 0.002 μm$^2$/s, avg. % immobile = 42.3%, n = 16) (*Figure 3—figure supplement 3*).

With the evidence of the nanoscale organization of AKAP150 and AC8 on the plasma membrane, we further hypothesized that the increased spatial density of Ca$^{2+}$-driven cAMP sources within the AKAP150 clusters, in conjunction with dispersed PDE1 in the cytosol (*Bender and Beavo, 2006*; *Goraya et al., 2008*), is important in compartmentalizing cAMP production and mediating the in-

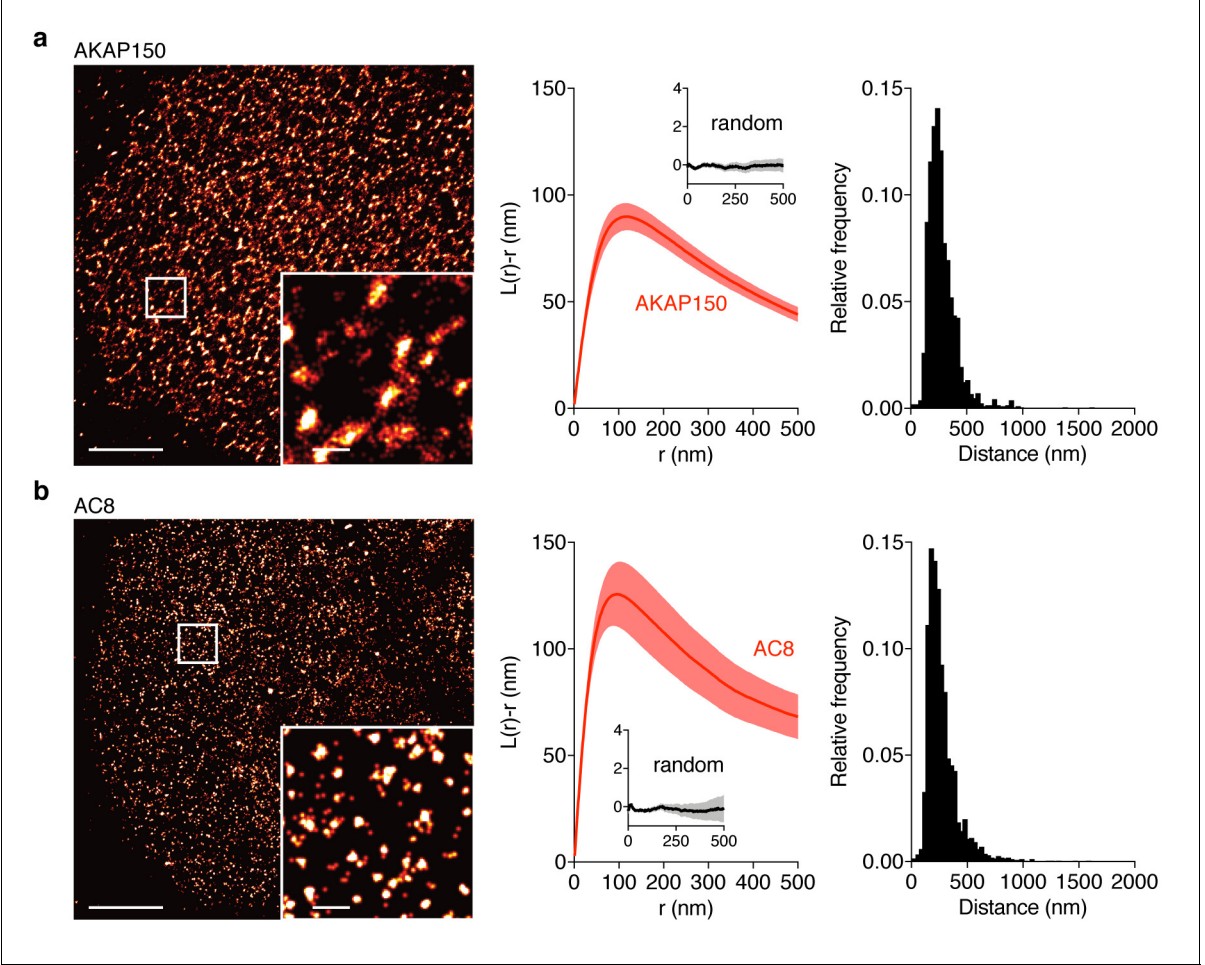

**Figure 3.** AKAP150 and AC8 both form nanoclusters at the surface of MIN6 β cells. (**a**) Representative super-resolution STORM image of the AKAP150 scaffold (scale 5 μm, inset 500 nm). Ripley-K analysis measures the average radii of the nanoclusters and indicates that AKAP150 forms clusters of 127 ± 9 nm, n = 20 (uniform random distribution control in inset). The nearest-neighbor distance distribution describes the distance between nanoclusters (average distance for AKAP150 is 313 ± 20 nm). (**b**) Representative super-resolution STORM image of $Ca^{2+}$-sensitive AC8 (scale 5 μm, inset 500 nm) depicts AC8 nanoclusters of average radius 88 ± 8 nm and average nearest-neighbor distance 292 ± 16 nm, n = 16.
The online version of this article includes the following figure supplement(s) for figure 3:

**Figure supplement 1.** Analysis of AKAP150 and AC8 cluster area.
**Figure supplement 2.** AKAP150 is co-clustered with AC8 at the plasma membrane in MIN6 cells.
**Figure supplement 3.** AC8 diffuses slowly at the plasma membrane.

phase cAMP signal. To test this idea, we sought to build a mathematical framework to describe the spatial compartmentalization of the in- and out-of-phase cAMP-$Ca^{2+}$ oscillations. Briefly, we expanded our network motif model (*Figure 2c*) by including a 3D spatial component with cAMP diffusion ($D_{cAMP}$ = 60 μm²/s; *Agarwal et al., 2016*) and incorporating our previously published well-mixed β cell model (*Ni et al., 2011*). We used the AKAP79/150:AC8 cluster pattern measurements from the STORM imaging to set model parameters in a hexagonal prism domain (200 nm edge, 600 nm depth), with one immobile AKAP79/150:AC8 cluster centered in the domain for simulation (*Figure 4a*, see Appendix 1 for model development details). By localizing AC8 within the AKAP79/150:AC8 cluster on the plasma membrane face and leaving PDE1 well-mixed throughout the volume, we could simulate $Ca^{2+}$-driven cAMP oscillations that were in-phase within the immediate vicinity of a cluster, but sharply transitioned out-of-phase outside the cluster. Specifically, during a $Ca^{2+}$ influx

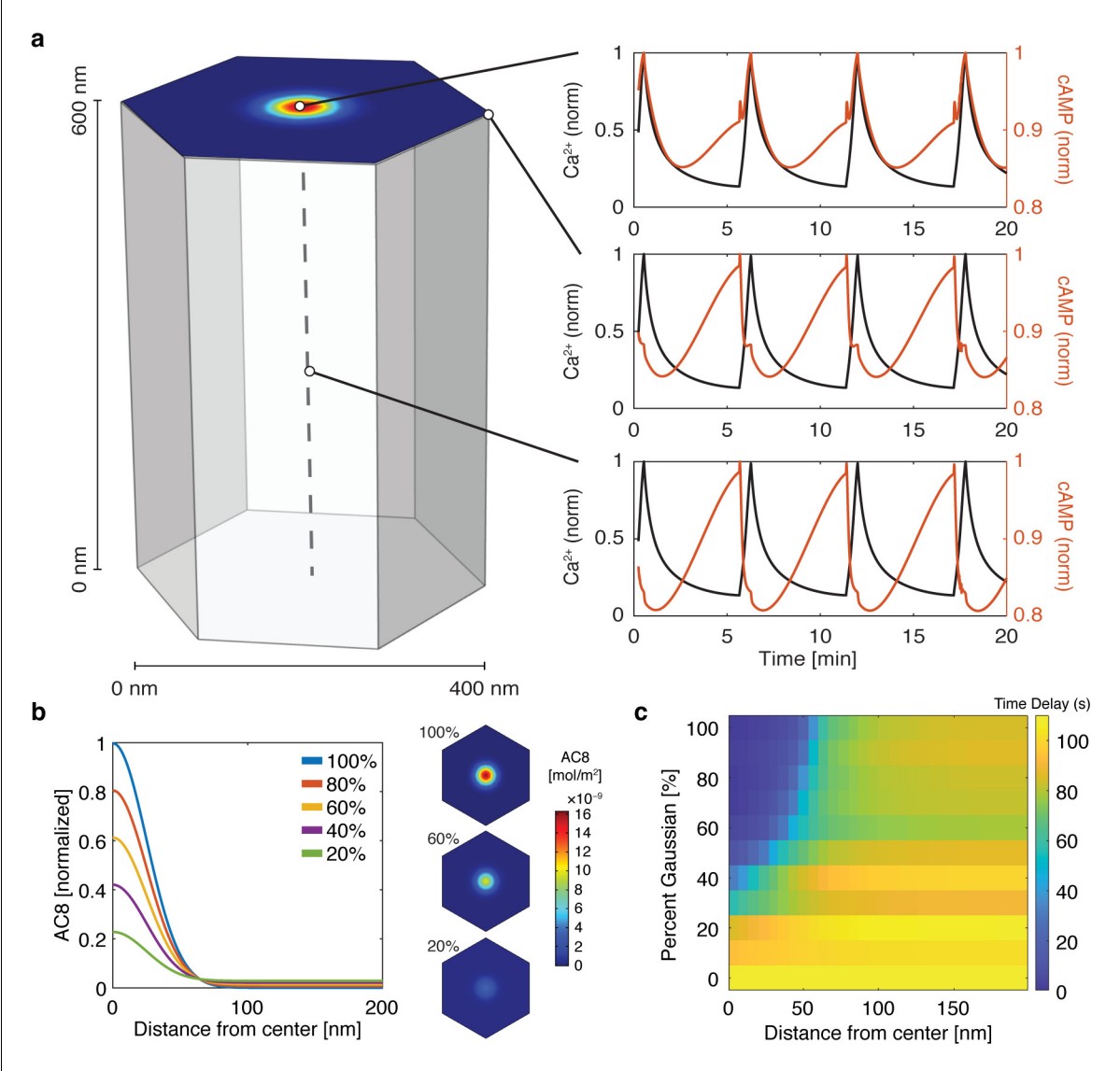

**Figure 4.** cAMP-Ca$^{2+}$ phase relationship can be described by a 3D reaction-diffusion model involving clusters of AKAP79/150 and AC8. (**a**) 3D reaction-diffusion model with a single AKAP79/150:AC8 co-cluster positioned at the PM in the β cell in a hexagonal prism volume. cAMP oscillates in-phase immediately within the AKAP79/150:AC8 nanocluster due to the high effective concentration of AC8, but out-of-phase at the PM or cytosol due to the presence of PDE1 (cAMP – red, Ca$^{2+}$– blue). (**b**) Disruption of the AKAP79/150:AC8 interaction can redistribute AC8 from within the cluster to the PM, shown by the half-Gaussian cross-sections and representative AC8 concentration heatmaps at the PM. (**c**) Heatmap depicting the time lag (s) for AC8 distribution (% Gaussian) and spatial distance (nm) from cluster center along PM.

The online version of this article includes the following figure supplement(s) for figure 4:

**Figure supplement 1.** Increasing AC8 concentration within the cluster leads to loss of compartmentalized cAMP-calcium phase relationship.

event, Ca$^{2+}$-triggered cAMP production dominated at the center of the AKAP79/150 cluster while Ca$^{2+}$-triggered cAMP degradation was favored outside the cluster at the PM and in the center of the unit volume (*Figure 4a*). Not surprisingly, the regime that recapitulates this phase relationship is sensitive to the spatially-restricted AC8/PDE1 balance and the diffusivity of cAMP. For example, by increasing the AC8 concentration within the cluster, the Ca$^{2+}$-driven cAMP generative flux could exceed the rate of cAMP degradation by PDE1 and thus cAMP and Ca$^{2+}$ oscillations could exist in-phase within both compartments (*Figure 4—figure supplement 1*). Alternatively, assuming that AC8 clustering is driven by AKAP150:AC8 interactions, weakening this interaction would then reduce the AC8 cluster stabilization and lead to a redistribution of AC8 away from the nanoclusters and a

decrease in the local concentration of AC8 within the clusters (*Figure 4b*). Without the high local concentration of AC8 driving a net positive cAMP production within an AKAP79/150 cluster, the spatial domain where cAMP oscillates in-phase with $Ca^{2+}$ is predicted to shrink while the out-of-phase regime expands and can reverse the phase at the cluster center (*Figure 4c*).

To test this prediction, we overexpressed the amino terminus of AC8 (AC8[1-106]) required for interaction with AKAP79/150 (*Willoughby et al., 2010*) to compete with the binding of endogenous AC8 to the endogenous AKAP150 scaffold. The disruption of the AKAP150:AC8 interaction was validated using a proximity ligation assay (PLA) to visualize the AKAP150:AC8 interaction in situ. Compared to non-transfected cells, cells expressing the AC8[1-106] peptide had a 39 ± 4% reduction in the PLA signals, indicating a decrease in the AKAP150:AC8 interaction (*Figure 5—figure supplement 1a,b*). Furthermore, STORM imaging showed that overexpression of the AC8[1-106] peptide led to a decrease in the percentage of AC8 single-molecule localizations within AC8 nanoclusters (n = 9) (*Figure 5a*), consistent with the predicted redistribution of AC8 molecules (*Figure 4b*). To test the impact of loss of AC8 molecules from the nanoclusters on the oscillation phase, we measured AKAP79/150-localized cAMP in the presence of AC8[1-106] and observed a significant increase in the average lag time (43 ± 6 s, n = 33) (*Figure 5b*). This is due to a higher proportion of cells exhibiting out-of-phase cAMP oscillations, indicating that the AKAP79/150:AC8 competitor peptide was

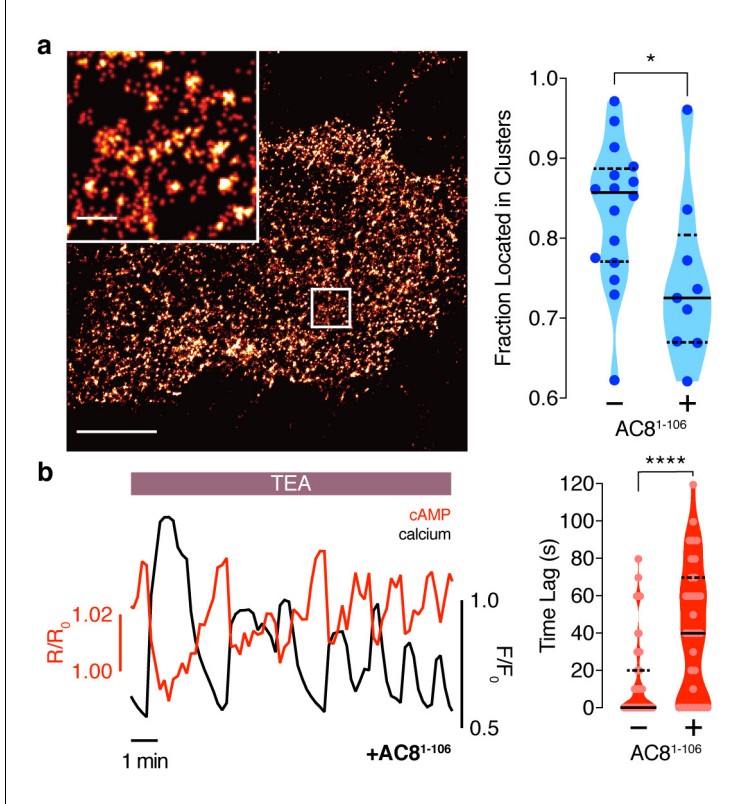

**Figure 5.** Disruption of the AKAP79/150:AC8 interaction is associated with a redistribution of AC8 at the PM and a phase shift of cAMP at the AKAP79/150 nanodomain. (**a**) Over-expression of the N-terminus of AC8, which is necessary and sufficient for mediating the AKAP79/150:AC8 interaction, redistributes AC8 from within nanoclusters to the general PM, as seen in the STORM image (scale 5 μm, inset 500 nm) and measured by the percent of localizations that fall into nanoclusters. (**b**) Disruption of the AKAP79/150:AC8 interaction lengthens the time lag between the cAMP (red) and $Ca^{2+}$ (black) signals at the AKAP79/150 compartment (avg. time lag in absence of disruptor is 13 ± 3 s, n = 60, and in presence of disruptor 43 ± 6 s, n = 33) due to more cells displaying out-of-phase cAMP oscillations. *p=0.0341, ****p<0.0001; unpaired two-tailed Students t-test.

The online version of this article includes the following figure supplement(s) for figure 5:

**Figure supplement 1.** Expression of the N-terminus of AC8 perturbed the interaction between endogenous AKAP150 and AC8 in MIN6.

sufficient in reversing the phase relationship in the AKAP79/150 compartment. Interestingly, we also observed many cells displaying irregular $Ca^{2+}$ oscillations, as indicated by a disruption in the periodic timing of individual cells' $Ca^{2+}$ peaks (*Figure 5b*, left). This nanoscale perturbation establishes the regulatory role of the AKAP79/150:AC8 interaction in mediating the compartmentalized cAMP-$Ca^{2+}$ phase relationship.

## AKAP79/150-mediated phase relationship is critical for regulating oscillatory $Ca^{2+}$

Next we systematically examined the impact of perturbing the precisely regulated phase relationship within the AKPA79/150 compartment. Due to the modulatory role of PKA in the $Ca^{2+}$-cAMP-PKA oscillatory circuit and the interaction between PKA and AKAP79/150, we wondered how the in-phase cAMP oscillations with respect to $Ca^{2+}$ are translated into PKA activities and if spatial compartmentalization of the phase relationship is also maintained at the PKA activity level. Therefore, we extended our 3D model to include AKAP79/150-associated PKA (see Appendix 1 for model details). In this extended model, PKA activity oscillations exhibit distinct phase relationships with respect to $Ca^{2+}$ within and outside of the AKAP79/150 compartment (*Figure 6—figure supplement 1a*). To test this prediction, we fused our FRET-based biosensor for PKA activity (AKAR4) (*Depry et al., 2011*) to either full-length AKAP79 or the PM-targeting motif and expressed the sensors in MIN6 cells. Upon TEA stimulation, PKA activity was observed to oscillate with a lag time of $25 \pm 6$ s (n = 15) within the AKAP79/150 compartment but with a lag time of $55 \pm 8$ s (n = 12) (*Figure 6—figure supplement 1b–e*) at the general plasma membrane, indicating that the compartmentalized phase relationship is preserved from cAMP to PKA.

Spatiotemporal organization of PKA signaling and its phosphorylation targets via AKAPs have been implicated in regulating several important pathways. For example, PKA has been shown to phosphorylate $Ca_V1.2$ in an AKAP79/150-dependent manner, and this modification can influence the open probability of the channel (*Murphy et al., 2014*), suggesting a mechanistic link between local cAMP/PKA activity and global oscillatory $Ca^{2+}$. Thus, we sought to study the functional role of the spatially compartmentalized cAMP-$Ca^{2+}$ phase relationship in regulating intracellular $Ca^{2+}$ dynamics. We measured $Ca^{2+}$ oscillations by RCaMP in the presence of either the EGFP-tagged AKAP79/150: AC8 disruptor peptide, $AC8^{1-106}$, or EGFP alone as a control. Population-wide differences in $Ca^{2+}$ dynamics, such as strength and timing, were observed in $AC8^{1-106}$-transfected cells and visualized in heat maps depicting the normalized $Ca^{2+}$ signal per cell versus time (*Figure 6a*). Interestingly, we found that the expression of the disruptor peptide was correlated with a significant decrease in the peak ratio between the second $Ca^{2+}$ peak and the first $Ca^{2+}$ peak (control average $-1.6\%$, n = 270; $AC8^{1-108}$ average $-10.8\%$, n = 562) post TEA addition, indicative of less sustained oscillations (*Figure 6b,c*). In addition to intracellular $Ca^{2+}$ concentration, the precise timing of internal oscillatory events is critical for modulating β cell functions such as glucose homeostasis and pulsatile insulin secretion (*Fridlyand et al., 2010*). In the presence of the disruptor peptide, cells also exhibited a longer elapsed time between oscillatory $Ca^{2+}$ peaks (control average $3.9 \pm 0.1$ min, n = 270; $AC8^{1-108}$ average $4.6 \pm 0.1$ min, n = 562), suggesting that the timing of the signaling circuit was disturbed (*Figure 6b,c*). It is well established that besides the precise timing, the regularity of cytoplasmic $Ca^{2+}$ in β cells is crucial in mediating pulsatile insulin secretion from the pancreas (*Gilon et al., 2002*; *Schmitz et al., 2002*). By stratifying the disruptor peptide-expressing cell population into 'low,' 'medium,' and 'high' expressers and performing a blinded classification of responding cells based on the regularity of the $Ca^{2+}$ oscillation (see Appendix 1 for details), we found a positive correlation between the percentage of cells exhibiting irregular oscillations and the expression level of the disruptor peptide (42% for low-expressing vs. 68% for high-expressing $AC8^{1-106}$ disruptor) (*Figure 6d*). Taken together, these data signify that the compartmentalized cAMP-$Ca^{2+}$ phase relationship regulates the oscillatory $Ca^{2+}$ signal and plays an important role in determining the pace, regularity, and sustainability of the $Ca^{2+}$ oscillations.

## Discussion

Biological oscillations represent a rich way of encoding information using the amplitude and frequency. Here, we show the phase in an oscillatory signaling circuit represents a novel mode of informational encoding for which the phase itself can be spatiotemporally regulated. In the case of the

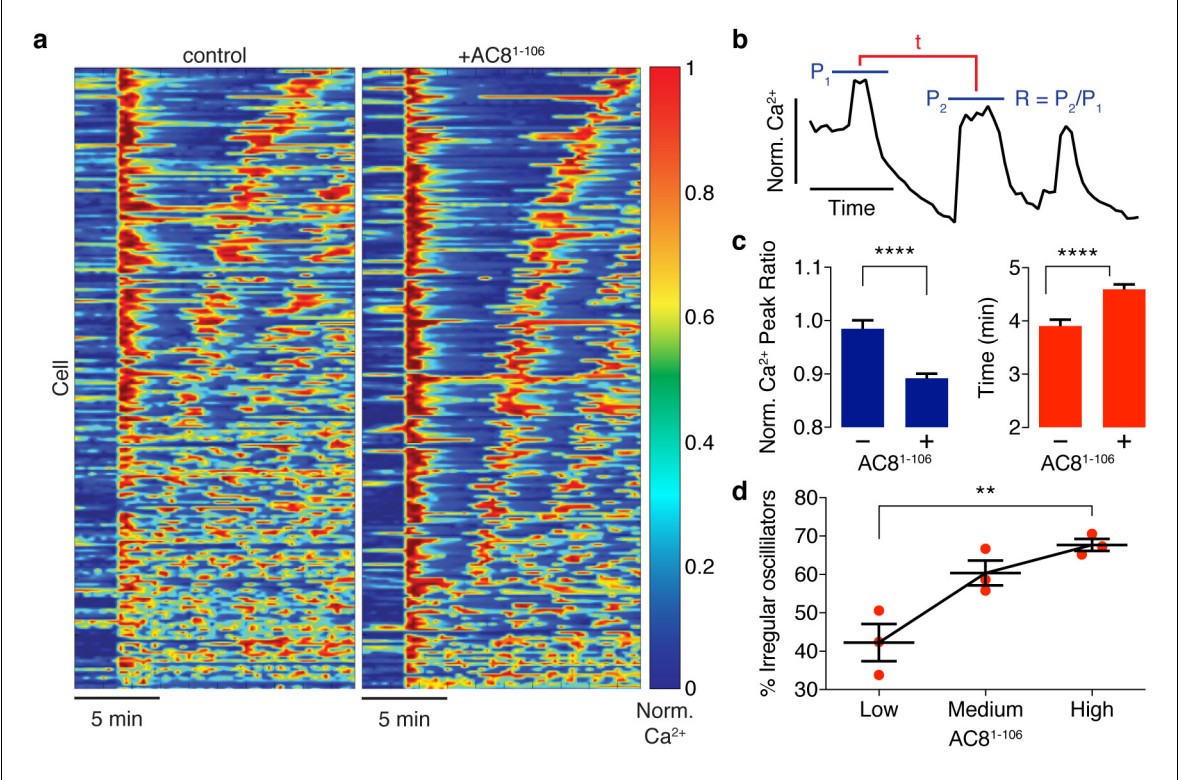

**Figure 6.** $Ca^{2+}$ oscillatory dynamics are affected by expression of the disruptor peptide in β cells. (**a**) Heatmaps depicting $Ca^{2+}$ oscillations for 220 randomly selected cells co-expressing EGFP alone (control, left) or EGFP-tagged AC8[1-106] (AKAP79/150:AC8 disruptor, right), ordered by a mixed parameter describing the time lag between the first two $Ca^{2+}$ peaks and the avg. timelag between all $Ca^{2+}$ peaks. (**b**) Schematic describing two $Ca^{2+}$ oscillatory parameters: the ratio between the first two $Ca^{2+}$ peaks ($R = P_2 / P_1$) and the interpeak timing (t). (**c**) The peak ratio R is decreased in the presence of the AKAP79/150:AC8 disruptor (left), indicating less of a sustained $Ca^{2+}$ oscillatory response. Over-expression of the disruptor also lengthens the timing between peaks (right). (**d**) The level of disruptor peptide expression is correlated with an increase in the percentage of cells exhibiting irregular oscillations (total n = 562). p=0.0054, ordinary one-way ANOVA.

The online version of this article includes the following figure supplement(s) for figure 6:

**Figure supplement 1.** The phase of oscillating PKA activity, relative to $Ca^{2+}$, is also spatially compartmentalized.

**Figure supplement 2.** Stochastic model results for the simplified model show the cAMP-$Ca^{2+}$ phase relationship is preserved and noise is reduced in higher flux conditions.

$Ca^{2+}$-cAMP-PKA circuit, the oscillatory cAMP/PKA phase relative to a widespread $Ca^{2+}$ signal is distinctly regulated within two adjacent membrane compartments through intracellular organization of scaffolds and signaling effectors. Localized perturbation of this spatial phase signature disrupts global $Ca^{2+}$ oscillations and thus has far-reaching consequences on the functional landscape of the β cell.

Compartmentalization of cAMP/PKA signaling is instrumental in processing a diverse set of inputs and mediating specific cellular functions; however, the mechanistic details of compartmentalization await to be fully elucidated (*Musheshe et al., 2018*). Given the measured kinetic rates of most ACs and PDEs, the generation of local cAMP gradients around single enzymes appears unfeasible (*Conti et al., 2014*). Context-dependent discrepancies in some of the kinetics (i.e. differences between in vitro and in vivo measurements) and slower cAMP diffusion due to buffering have been shown to be important for cAMP compartmentalization (*Bock et al., 2020*; *Zhang et al., 2020*). Here, we propose the nanoscale organization of key cAMP effectors and regulators as a novel mechanism for cAMP compartmentation. Despite the slow rates measured for individual ACs, we computationally and experimentally describe conditions in which the generation of compartmentalized

cAMP can emerge from the clustering of many relatively immobile AC8 enzymes at the membrane and bulk distribution of PDE1 in the cytoplasm. Alterations to this nanoscale organization lead to dysregulated $Ca^{2+}$ oscillations, demonstrating the functional importance of maintaining this organization. This system also serves as a nanoscale demonstration of how a cell can translate a global signal ($Ca^{2+}$) into a compartmentalized signal (cAMP/PKA activity) by local activation and global inhibition, a strategy that is likely utilized in many other cellular contexts (*Levchenko et al., 2000*; *Purvis and Lahav, 2013*).

In order to understand the functional impact of the nanoscale architecture, it is important to also consider the discrete scale of such signaling organization. According to our rough estimates, approximately 15–150 individual AC8 molecules comprise AKAP150:AC8 nanoclusters (see Materials and methods for STORM-based estimation). To investigate the role of small molecule numbers, we recast our previous network motif model as a stochastic system and simulated the impact of changing initial conditions of PDE1 and AC8. We found that even in these discrete simulations, the previously noted phase relationships are preserved. As the balance between PDE and AC is perturbed, the system moves from an in-phase to an out-of-phase cAMP response with respect to $Ca^{2+}$. Interestingly, we also noted that increasing the number of AC8 molecules leads to a reduction of the noise in the cAMP response. Thus, we predict that tuning AC8 cluster size might be a biological design strategy of nanoclusters to maintain accurate responses to perturbations (*Figure 6—figure supplement 2a–d*).

The interplay between $Ca^{2+}$ and cAMP is very intricate, and multiple mechanisms could contribute to the observed dysregulated signaling effects upon perturbation of the AKAP79/150-compartmentalized cAMP-$Ca^{2+}$ phase (*Figure 6a–d*). AKAPs can recruit PKA to regulate channel activities (*Dell'Acqua et al., 2006*; *Mo et al., 2017*; *Torres-Quesada et al., 2017*), such as in the regulation of voltage-mediated $Ca^{2+}$ entry via PKA-dependent phosphorylation of $Ca_V1.2$ (*Murphy et al., 2014*) or the modulation of store-operated $Ca^{2+}$ entry by both PKA-dependent STIM1 and Orai1 phosphorylation (*Thompson and Shuttleworth, 2015*; *Zhang et al., 2019*). Additional levels of regulatory feedback within the $Ca^{2+}$-cAMP-PKA oscillatory circuit have also been identified, such as a negative feedback loop involving PKA phosphorylation of AC8, thereby fine-tuning the circuit dynamics (*Willoughby et al., 2012*).

The precisely regulated $Ca^{2+}$ and cAMP oscillations are likely functionally important as insulin secretion requires coordination between $Ca^{2+}$, cAMP, and PKA. Localized cAMP/PKA signaling at the AKAP79/150 scaffold and the in-phase oscillations of cAMP and $Ca^{2+}$ within this compartment might play a critical role in regulating insulin secretion due to close interactions between AKAP79/150 and insulin secretory granules via $Ca_V1.2$ (*Barg et al., 2001*). Several important processes and components of the secretory machinery have been identified as targets of PKA signaling here, such as PKA-dependent mobilization of granules (*Renström et al., 1997*) and modulation of the synaptosomal protein SNAP25 (*Gao et al., 2016*). These PKA phosphorylation events may be highly coordinated with oscillations of $Ca^{2+}$, which drive the exocytosis process for pulsatile insulin secretion. In addition to PKA-dependent secretory control, cAMP has recently been implicated to play a direct role in fusion pore formation via the cAMP-regulated guanine exchange factor Epac (*Guček et al., 2019*). Compartmentalized phase regulation of the $Ca^{2+}$-cAMP-PKA oscillatory circuit at the AKAP79/150 macromolecular complex is likely involved in the regulation of many β cell processes, in addition to insulin secretion, and more work will be needed to further establish the mechanisms involved in decoding the information embedded in the local phase relationship.

The $Ca^{2+}$-cAMP-PKA oscillatory circuit in pancreatic β cells integrates many important regulators of cellular function, and the precise coordination of each is required for proper signaling control. Here, we have uncovered a spatiotemporal organization of the circuit where the oscillatory phase between cAMP/PKA and $Ca^{2+}$ depends on the spatial proximity of the AKAP79/150 scaffold protein and AC8. The construction principles of this signaling nanodomain, including the spatial distributions of sinks and sources, likely represent a generalized strategy for the generation of other compartmentalized signals and provide a unique modality in which cells embed, process, and produce signaling information.

# Materials and methods

## Key resources table

| Reagent type (species) or resource | Designation | Source or reference | Identifiers | Additional information |
|---|---|---|---|---|
| Cell line (*Mus musculus)* | MIN6 | Dr. Jun-Ichi Miyazaki, Osaka University | RRID:CVCL_0431 | |
| Chemical compound, drug | Tetraethylammonium chloride (TEA) | Sigma | T2265 | 20 mM |
| Chemical compound, drug | 8-Methoxymethyl-3-isobutyl-1-methylzanthine (8MM-IBMX) | Sigma-Aldrich | M2547 | 100 µM |
| Chemical compound, drug | Milrinone | Alexis | Cat# ALX-270–083 M005 | 10 µM |
| Chemical compound, drug | Rolipram | Alexis | Cat# ALX-270–119 | 1 µM |
| Chemical compound, Drug | KCl | Sigma-Aldrich | P9541 | 15 mM |
| Commercial assay or kit | Lipofectamine-2000 | Invitrogen | 11668019 | |
| Recombinant DNA reagent | pCDNA3 AKAR4 | PMID:20838685 | | |
| Recombinant DNA reagent | (Ci/Ce)Epac2-camps | Dr. Dermot Cooper, University of Cambridge | | |
| Recombinant DNA reagent | AKAP79 (*AKAP5, Homo sapiens*) | Dr. John D. Scott, University of Washington | | |
| Recombinant DNA reagent | AC8 (*Adcy8, Rattus norvegicus*) | Dr. Dermot Cooper, University of Cambridge | | |
| Recombinant DNA reagent | *ShAdcy8* Plasmid | Dr. Jochen Lang, University of Bordeaux | *ShAdcy8* #2 | |
| Recombinant DNA reagent | RCaMP | Dr. Loren Looger, Janelia Farms | | |
| Recombinant DNA reagent | R-FlincA | Dr. Kazuki Horikawa, Tokushima University Graduate School | | |
| Commercial assay or kit | Duolink in situ red starter kit | Sigma | DUO92101 | |
| Commercial assay or kit | Duolink in situ Probemaker MINUS | Sigma | DUO92010 | |
| Commercial assay or kit | Duolink in situ Probemaker PLUS | Sigma | DUO92009 | |
| Antibody | Anti-AC8, rat polyclonal | Abcam | ab196686 | 1:2000 |
| Antibody | Anti-AKAP150, rat polyclonal | Millipore Sigma | 07–210 | 1:500 |
| Antibody | Anti-AKAP79, mouse monoclonal | BD | 610314 | 1:500 |
| Antibody | Goat anti-rabbit AlexaFluor647 | ThermoFisher Scientific | A21245 | 1:1000 |
| Antibody | Goat anti-mouse AlexaFluor568 | ThermoFisher Scientific | A11031 | 1:1000 |
| Software, algorithm | FIJI | https://imagej.net/Fiji | RRID:SCR_014294 | |
| Software, algorithm | MetaFluor | https://www.molecular devices.com/ | RRID:SCR_002285 | |
| Software, algorithm | MATLAB | https://www.mathworks.com/ | RRID:SCR_001622 | |

*Continued on next page*

*Continued*

| Reagent type (species) or reagent | Designation | Source or reference | Identifiers | Additional information |
|---|---|---|---|---|
| Software, algorithm | GraphPad Prism | https://www.graphpad.com/scientific-software/prism/ | RRID:SCR_002798 | |
| Software, algorithm | COPASI | http://copasi.org/ *Hoops et al., 2006* | RRID:SCR_014260 | |
| Software, algorithm | Virtual Cell | https://vcell.org/ *Cowan et al., 2012* | RRID:SCR_007421 | |

## Gene construction

For AKAP79-(Ci/Ce)Epac2-camps, AKAP79 (gene *AKAP5*, from Dr. John D. Scott) was PCR amplified to have HindIII/BamHI digestion sites, and (Ci/Ce)Epac2-camps (from Dr. Dermot Cooper) was PCR amplified to have BamHI/EcoRI digestions sites. Both fragments were inserted into a pcDNA3 (Invitrogen) backbone for mammalian expression (cAMP sensor is C-terminal to AKAP79). For AKAP79-AKAR4, a similar approach was taken where AKAR4 was dropped between BamHI/EcoRI. For AC8 (gene *Adcy8*, from Dr. D. Cooper), AC8[1-108], Gibson Assembly was used to insert the genes into the pcDNA3 mammalian expression vector. The *ShAdcy8* construct for *Adcy8* knockdown was previously verified and a gift from Dr. Jochen Lang. RCaMP was a gift from Dr. Loren Looger.

## Cell culture

MIN6 cells (a mouse insulinoma β cell line, gift from Dr. Jun-Ichi Miyazaki) were plated onto sterilized glass coverslips in 35 mm dishes and grown to 50–90% confluency in DMEM (10% FBS, 4.5 g/L glucose) at 37°C with 5% $CO_2$. Identity was verified by phenotype ($Ca^{2+}$ oscillations, morphology, and insulin secretion). Recurrent mycoplasma testing was performed and results were negative. Cells were transfected using Lipofectamine 2000 (Invitrogen) for 20–48 hr before imaging.

## Imaging

Cells were washed twice with Hank's balanced salt solution buffer and maintained in the dark at room temperature. Cells were imaged on a Zeiss Axiovert 200M microscope with a cooled charge-coupled device camera (MicroMAX BFT512, Roper Scientific, Trenton, NJ) controlled by META-FLUOR 6.2 software (Universal Imaging, Downingtown, PA). Dual cyan/yellow emission from FRET used a 420DF20 excitation filter, a 450DRLP dichroic mirror, and two emission filters [475DF40 for CFP and 535DF25 for YFP]. RFP fluorescence was imaged using a 568D55 excitation filter, a 600DRLP dichroic mirror, and a 653DF95 emission filter. GFP fluorescence was imaged using a 480DF30 excitation filter, a 505DRLP dichroic mirror, and a 535DF45 emission filter. YFP fluorescence was imaged using a 495DF40 excitation filter, a 515DRLP dichroic mirror, and a 535DF25 emission filter. These filters were alternated by a filter-changer Lambda 10–2 (Sutter Instruments, Novato, CA). Exposure time was 50–500 ms, and images were taken every 10–30 s.

For confocal imaging, images were collected with a C2 plus on a Nikon Ti2 inverted microscope equipped with a Plan Apo lambda 60x oil immersion objective NA 1.4. YFP fluorescence was excited with the 488 nm line from a LU-N4 laser. Images were acquired with a DUVB detector collecting emission from 495 nm to 600 nm with a virtual spectral GaAsP detector controlled by NIS Elements software. The pinhole was set at 30 μm. Frame size was 1024 × 1024 pix.

For FRAP acquisition, cells were imaged using a C2 plus mounted on a Nikon Ti2 with a 488 nm laser (LU-N4 laser engine, Nikon Instruments) and sensitivity PMT detector (C2-DU3, Nikon) with 525/50 nm bandpass filter. The excitation light is reflected by a dichroic mirror 405/488/561/640 and focused through an Apo 100 × 1.49 NA objective. Image stacks were acquired in Galvano mode with unidirectional scanning with a 488 nm laser at 1.5% laser power with frame size 512 × 512 at scan zoom 2, one frame per second and 97.1 μm pinhole size. Small stimulation ROIs were drawn at the plasma membrane. The total FRAP series contained three images before bleaching (obtained with 2 s intervals), two cycles of ROI bleaching with the 488 nm laser at 100% laser power (5 frames at one frame per second), and two minutes of continuous acquisition to monitor fluorescence recovery.

## Image analysis

Fluorescence images were background-corrected by subtracting the fluorescence intensity of background with no cells from the emission intensities of cells expressing fluorescent reporters. Cells for experiments using the targeted biosensors were manually segmented from microscopy images based on emission intensity (YFP) and RCaMP (RFP) using custom-written Java and MATLAB scripts and whole-cell fluorescence emission was measured. The ratio of yellow/cyan emission (AKAR4) or cyan/yellow emission (Epac2-camps) and RFP intensity were then calculated at different time points using MATLAB scripts. The values of all-time courses were normalized by dividing each by the average basal value before drug addition. FRAP measurements were background-subtracted, bleach-corrected, and normalized, and fit to a previously described diffusion-dominant model (*Ellenberg et al., 1997*; *Phair et al., 2004*). For the AKAP150:AC8 disruption experiments, cells were segmented with custom CellProfiler (Broad Institute) pipelines based on RCaMP (RFP) fluorescence emission, and both RFP and GFP fluorescence intensities (EGFP-AC8$^{1-106}$) per cell were measured every 30 s. Graphpad Prism eight was used for visualization and statistical analysis. Custom-written MATLAB scripts were used for correlation analysis and regularity classification.

## Super-resolution imaging (STORM)

For fixed-cell stochastic optical reconstruction microscopy (STORM) imaging, cells were fixed with 4% paraformaldehyde (PFA) and 0.2% glutaraldehyde (GA) for 20 min and then washed with 100 mM glycine in HBSS to quench the free PFA. Cells were permeabilized and blocked in a permeabilization solution with 0.1% Triton X-100, 0.2% bovine serum albumin, 5% goat serum, and 0.01% sodium azide in HBSS. The cells were then incubated overnight at 4°C with an anti-AC8 antibody (Abcam, ab196686) at a 1:2000 dilution or an anti-AKAP150 (Millipore Sigma, 07–210) antibody at a 1:500 dilution, followed by 1 to 2 hr with goat anti-rabbit Alexa 647–conjugated antibody (Thermo-Fisher Scientific, A21245) at 1:1000 dilution. For the 2-color STORM experiments, cells were incubated with both the anti-AC8 antibody (1:2000 dilution) and an anti-AKAP79 antibody that recognizes a conserved epitope in AKAP150 (BD, 610314; raised against the C-terminal 248 residues of AKAP79 which shares 70% similarity with a region in AKAP150; 1:500 dilution), followed by a 2 hr incubation with a goat anti-rabbit Alexa 647-conjugated antibody (1:1000 dilution) and a goat anti-mouse Alexa 568-conjugated antibody (ThermoFisher Scientific, A11031; 1:1000 dilution). The cells were then post-fixed again in 4% PFA and 0.2% GA, quenched with 100 mM glycine in HBSS, and washed with HBSS to prepare for imaging. Immediately before imaging, the medium was changed to STORM-compatible buffer (50 mM Tris-HCl, pH 8.0, 10 mM NaCl, and 10% glucose) with glucose oxidase (560 µg/ml), catalase (170 µg/ml), and mercapto-ethylamide (7.7 mg/ml). STORM images were obtained using a Nikon Ti total internal reflection fluorescence (TIRF) microscope with N-STORM, an Andor IXON3 Ultra DU897 EMCCD, and a 100 × oil immersion TIRF objective. Photo-activation was driven by a Coherent 405 nm laser, while excitation was driven with a Coherent 647 nm laser or Coherent 561 nm laser.

All image analysis and image reconstruction were performed using both Nikon Elements analysis software and custom-written MATLAB scripts. Blinking correction was performed by implementing the pairwise Distance Distribution Correction (DDC) algorithm (*Bohrer et al., 2019*). Cluster property measurements were performed using Ripley-K analysis and custom mean-shift MATLAB code for segmentation, as described before (*Mo et al., 2017*). Co-clustering measurements were made using custom MATLAB scripts implementing Getis-Franklin pattern analysis, as described previously (*Getis and Franklin, 1987*; *Mo et al., 2017*). Briefly, we first calculated the degree of local clustering of AKAP150 molecules using the L(r) spatial statistic at r = 200 nm which counts the number of AKAP150 localizations within 200 nm (length scale of nanoclusters) of each AKAP150 localization, normalized appropriately. We then calculated Lcross(r) at r = 200 nm which counts the number of AC8 localizations within 200 nm of each AKAP150 localization, normalized appropriately. Plots of Lcross(200) vs. L(200) were generated and an L threshold of 150 was chosen based on the proportion of localizations found within clusters from the mean-shift segmentation analysis, and used to divide the plot into quadrants to calculate the proportion of AKAP150 localizations co-clustered with AC8 as previously described (*Mo et al., 2017*).

We obtained a very rough estimate of AC8 molecules within nanoclusters from our blink-corrected STORM datasets by assuming background localizations outside the cell are due to single

secondary antibodies with conjugated dyes. We measured an average of 2 localizations per secondary antibody outside the cell and approximately 50 localizations per AC8 nanocluster within the cell. Estimating a labeling stoichiometry of 0.4–1.5 secondary antibodies per primary antibody and 0.4–1.5 primary antibodies per AC8 molecule (*Zanacchi et al., 2017*; *Nieuwenhuizen et al., 2015*), we can roughly approximate the number of AC8 molecules per nanocluster to be between 15 and 150 molecules/cluster (N = 3 cells).

### Proximity ligation assay

Antibodies for AC8 and AKAP150, mentioned in STORM section, were buffer exchanged into DPBS and conjugated with MINUS or PLUS oligos, following the Sigma DuoLink in situ Probemaker kits. PLA experiments were performed using the Duolink in situ red kit for PLAs according to the provided protocol. The only protocol modification was to extend the amplification time by 50 min. Briefly, cells were fixed and permeabilized as in the STORM experiments before incubation with PLUS and MINUS oligo-conjugated primary antibodies for 30 min at 37˚C each, with washes after each step. Ligation of the nucleotides and amplification of the strand occurred sequentially by incubating cells with first ligase then polymerase and detection solution. PLA experiments with AKAP95 antibodies from different species were used as positive controls in HEK293T cells, and experiments with just one oligo-labeled primary antibody or the other were our negative control. Images were acquired on a Nikon Ti Eclipse epifluorescence scope with z-control and maximum intensity projections were created. A cross section of the DAPI-stained nucleus (3.6–5 µm) was also acquired and the number of dots per cell was counted using the nucleus as reference.

### Statistical analysis

Statistics, such as means, SEMs, and comparisons were calculated using Graphpad Prism 8 software. Experiments comparing multiple conditions were analyzed using one-way ANOVA or unpaired t-tests. Statistical significance was assessed using $p < 0.05$ as a cutoff. Experiments were repeated >3 times. Individual cells with high expression of the AKAP79/150-fused biosensors were removed from analysis (see Appendix 1 for details).

## Acknowledgements

We thank Dr. John D Scott for providing AKAP79, Dr. Dermot Cooper for providing (Ci/Ce)Epac2-camps and AC8, Dr. Loren Looger for providing RCaMP, Dr. Kazuki Horikawa for providing R-FlincA and Dr. Jochen Lang for providing the shAC8 construct. We thank Dr. Susan Taylor for critical reading of the manuscript. We also thank the members of the Nikon Imaging Center at UCSD for help with fluorescence microscopy experiments, and the Neuroscience Imaging Center at the Moore's Cancer Center for help with the STORM acquisition and analysis. This work was supported by NIH R01 DK073368 and R35 CA197622 (to JZ), DOD AFOSR FA9550-18-1-0051 (to PR and JZ), ONR N00014-17-1-2628 (to PR), NIH R01 GM086447 and NSF MCB 1817551 (to JX), NIH T32GM007231 (to CHB), Johns Hopkins Discovery Award (to JX and JZ) and Hamilton Innovation Rsearch Award (to JX).

## Additional information

### Competing interests

Jie Xiao: Reviewing editor, *eLife*. The other authors declare that no competing interests exist.

### Funding

| Funder | Grant reference number | Author |
|---|---|---|
| National Institutes of Health | R01 DK073368 | Jin Zhang |
| U.S. Department of Defense | DOD AFOSR FA9550-18-1-0051 | Padmini Rangamani<br>Jin Zhang |
| Office of Naval Research | ONR N00014-17-1-2628 | Padmini Rangamani |

| National Institutes of Health | R35 CA197622 | Jin Zhang |
|---|---|---|
| National Institutes of Health | 5T32 GM007231 | Christopher H Bohrer |
| National Institutes of Health | R01 GM086447 | Jie Xiao |
| National Science Foundation | MCB 1817551 | Jie Xiao |
| Johns Hopkins University | Johns Hopkins Discovery Award | Jie Xiao Jin Zhang |
| Johns Hopkins University | Hamilton Innovation Research Award | Jie Xiao |

The funders had no role in study design, data collection and interpretation, or the decision to submit the work for publication.

## Author contributions

Brian Tenner, Conceptualization, Resources, Data curation, Formal analysis, Supervision, Validation, Investigation, Methodology, Writing - original draft, Project administration, Writing - review and editing; Michael Getz, Software, Formal analysis, Validation, Visualization, Writing - review and editing; Brian Ross, Data curation, Software, Investigation, Visualization, Methodology, Writing - review and editing; Donya Ohadi, Data curation, Software, Formal analysis, Validation, Writing - review and editing; Christopher H Bohrer, Data curation, Software, Formal analysis, Methodology, Writing - review and editing; Eric Greenwald, Conceptualization, Data curation, Software, Formal analysis, Writing - review and editing; Sohum Mehta, Conceptualization, Resources, Supervision, Investigation, Visualization, Project administration, Writing - review and editing; Jie Xiao, Project administration, Writing - review and editing; Padmini Rangamani, Conceptualization, Supervision, Investigation, Project administration, Writing - review and editing; Jin Zhang, Conceptualization, Supervision, Funding acquisition, Project administration, Writing - review and editing

## Author ORCIDs

Michael Getz ⬥ https://orcid.org/0000-0001-9886-8454
Brian Ross ⬥ http://orcid.org/0000-0001-9020-627X
Sohum Mehta ⬥ https://orcid.org/0000-0003-4764-8579
Jie Xiao ⬥ http://orcid.org/0000-0003-1433-5437
Padmini Rangamani ⬥ https://orcid.org/0000-0001-5953-4347
Jin Zhang ⬥ https://orcid.org/0000-0001-7145-7823

## Decision letter and Author response

Decision letter https://doi.org/10.7554/eLife.55013.sa1
Author response https://doi.org/10.7554/eLife.55013.sa2

# Additional files

## Supplementary files

• Transparent reporting form

## Data availability

Data has been deposited at Dryad (https://doi.org/10.6075/J0NP22TK). Analysis scripts have been deposited on GitHub (https://github.com/btenner/calcium-cAMP-PKA; copy archived at https://archive.softwareheritage.org/swh:1:rev:5c4c971e275cdc3e97b891e4b507393b209f80a1/).

The following dataset was generated:

| Author(s) | Year | Dataset title | Dataset URL | Database and Identifier |
|---|---|---|---|---|
| Tenner B | 2020 | Spatially compartmentalized phase regulation of a Ca2+-cAMP-PKA oscillatory circuit | https://doi.org/10.6075/J0NP22TK | Dryad Digital Repository, 10.6075/J0NP22TK |

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

## Appendix 1

### cAMP analysis and quantification

AKAP79-(Ci/Ce)Epac2-camps transfected cells displayed cAMP oscillations that were either in-phase or out-of-phase with their respective $Ca^{2+}$ signal. This was in sharp contrast to responsive lyn-(Ci/Ce) Epac2-camps transfected cells where all cells yielded only out-of-phase oscillations. We found a strong correlation between the AKAP79-fused sensor expression level and the observed cAMP-$Ca^{2+}$ phase relationship, with cells having lower levels of sensor present displaying predominantly in-phase cAMP oscillations and cells with higher levels of the AKAP79/150-fused biosensor exhibiting out-of-phase oscillations (*Appendix 1—figure 1*). Overexpression of the AKAP79 scaffold likely changed the stoichiometry of the signaling complexes and resulted in unsuccessful targeting of the biosensor to functional AKAP79/150 domains, and so in this manuscript we considered only TEA-responsive cells below an AKAP79 expression threshold determined by the YFP fluorescence (*Appendix 1—figure 1*).

### Time lag calculation

Due to the heterogeneity of cellular $Ca^{2+}$ and cAMP/PKA activity oscillatory responses in each cell (i.e. variations in frequency, amplitude, and regularity), we sought an applicable metric to describe the phase relationship. Here we measure the lag time (sec) between the $Ca^{2+}$ signal trace and the cAMP/PKA activity signal trace. Specifically, we high-pass filtered both the $Ca^{2+}$ and cAMP/PKA activity traces (approximately 20 min) to subtract out slowly varying baseline changes, normalized the traces so that the maximum intensity/FRET ratio was set to 1, and then computed the cross-correlation to measure the signal overlap for different lag times. To calculate the lag time, we identified peaks in the cross-correlation passing a peak prominence cutoff and found the absolute value of the shortest lag time corresponding to a peak maximum. For in-phase oscillations, the lag time was typically small ($\tau \leq 20$ s) due to the two signal traces oscillating in synchrony. On the other hand, out-of-phase oscillations typically corresponded to longer lag times ($\tau > 20$ s) due to the anti-phasic relationship seen in the peak timing and peak shape. Analysis was performed with custom scripts in MATLAB and Java, and pipelines in CellProfiler.

### Quantification of nanodomain perturbation effects on global $Ca^{2+}$

In order to measure the effects of AKAP79/150:AC8 disruption on $Ca^{2+}$ dynamics, we transiently transfected and expressed $AC8^{1\text{-}106}$ in MIN6 cells and measured $Ca^{2+}$ with RCaMP. For quantification of the interpeak timing and peak ratio, we first selected cells that responded to the TEA treatment, identified $Ca^{2+}$ peaks passing a peak prominence cutoff, and finally calculated the avg. time between peak maxima and RFP intensity ratio between the second and first $Ca^{2+}$ peak maxima. To find the percentage of cells with regular vs. irregular $Ca^{2+}$ oscillations, we randomized all $Ca^{2+}$ traces from the experimental and control samples and performed a blinded classification to sort the single cell traces as regular, irregular, or nonresponsive. Analysis was performed with custom scripts in MATLAB and Java, and pipelines in CellProfiler.

### Computational modeling
Well-mixed system
Assumptions

- Signaling components are present in large enough quantities that concentration changes are smooth and move in a deterministic fashion.
- Well-mixed kinetic rate constants contain conversion factors between compartments, that is membrane to cytosol.
- Binding interactions occur rapidly enough such that any kinetic parameter, *k*, remains constant on surfaces (*Berry, 2002*).
- AKAP does not alter the activity of the catalytic subunits of Protein kinase A (PKA) instead it only affects localization.

- $Ca^{2+}$ independent activity of Adenylyl cyclase (AC) is of the same strength as inactive $Ca^{2+}$ dependent Adenylyl cyclase (AC8) and stays at a constant value.

## Modeling well-mixed chemical reactions

### Mass-action kinetics

We generated an ordinary differential equation (ODE) for every species using mass-action kinetics for each binding interaction. The law of mass-action states that the rate of a chemical reaction is proportional to the product of the concentration of the reactants raised to the power of their stoichiometric coefficient (*Guldberg and Waage, 1879*). Mass-action kinetics rely on the assumption that the rate constant, $k$, is constant over time. For example, consider the one-reaction system:

$$A + 2B \rightleftharpoons 3C,$$

where the forward and backward rates are $k_1$ and $k_2$. The differential equations describing the dynamics of species $A, B$, and $C$ under mass-action kinetics are:

$$\frac{d[A]}{dt} = k_2[C]^3 - k_1[A][B]^2 \quad \frac{d[B]}{dt} = 2k_2[C]^3 - 2k_1[A][B]^2 \quad \frac{d[C]}{dt} = 3k_1[A][B]^2 - 3k_2[C]^3.$$

### Michaelis-Menten kinetics

We used Michaelis-Menten kinetics to model the kinetics of enzyme-catalyzed reactions. When a reaction is catalyzed by an enzyme with kinetic properties $k_{cat}$ and $K_M$,

$$S E \rightarrow P$$

then the reaction rate is given by

$$\frac{d[S]}{dt} = -\frac{k_{cat}[E][S]}{K_M + [S]} = \frac{d[P]}{dt}.$$

For Michaelis-Menten kinetics, concentrations of reactants and products must be in large enough quantities, and one of the following conditions must apply: the concentration of the substrate is much larger than the concentration of products, $[\text{S}] \gg [\text{p}]$, and/or the energy released in the reaction is very large, $\Delta G \ll 0$.

### Model development

We constructed a biochemical network to represent interactions between $Ca^{2+}$ and cAMP in $\beta$92-cells, *Figure 1b*, (*Fridlyand and Philipson, 2016*; *Ni et al., 2011*) after a depolarization event. The computational model considered the dynamics of calcium, potassium, leaky, and calcium-sensitive potassium channels (*Appendix 1—table 1*). Importantly, we included feedback of PKA with KATP channels and the inclusion of $Ca^{2+}$-sensitive ACs and PDEs. The model contains a total of **92** parameters with **11** free parameters. The values of the parameters are constrained through both previously peer-reviewed publication results (*Ang and Antoni, 2002*; *Boras et al., 2014*; *Lai et al., 2015*; *Masada et al., 2009*; *Ni et al., 2011*) and with new experimental results using obtained FRET measurements. To constrain source and sink activation rates, previously published literature of AC and PDE stimulus-response curves (of related isoforms) was utilized (*Ang and Antoni, 2002*; *Masada et al., 2009*). COPASI was used to calculate initial guesses for kinetic activation parameters. FRET measurements took precedence over binding curves, especially for the spatial model where further fitting routines were performed to refine the model. Model variations were performed to attain both semi-physiological concentrations (within the ranges of the sensor) and phase (period and relation) information. Predictions are made on qualitative behavior as opposed to quantitative as proper parameter fitting would require much more data than available for this system.

Values and reaction sets used in the well-mixed model can be found in *Appendix 1—tables 1–5*. The network of interactions was constructed using COPASI (version 4.23, build 184) (http://www.nrcam.uchc.edu, http://copasi.org/) (*Hoops et al., 2006*). The model was built in COPASI to leverage the inbuilt fitting techniques for initial parameter guesses pre-FRET.

## Well-mixed computational results

The network shown in *Figure 1b* has been shown to exhibit oscillations through cAMP variation due to the action of PKA on IP3 receptors and KATP plasma membrane channels. This network has been studied in many labs previous work (*Fridlyand et al., 2007*; *Fridlyand et al., 2003*; *Han et al., 1999*; *Ni et al., 2011*) and has been explored to show:

- The system can be moved in and out-of-phase through tuning of AC (*Fridlyand and Philipson, 2016*).
- The oscillation rate can be tuned through PKA feedback to KATP channels (*Ni et al., 2011*).
- Both $Ca^{2+}$ sensitive PDE and AC is necessary for oscillations to occur (*Ni et al., 2011*).

Our model results agree with the above findings.

Variations in the connection strength of sources and sinks also introduced another finding—changing the component's (source or sink) variability in the regime of $Ca^{2+}$ spiking (0.1-1.2 μ$M$) a switch in-phase can also be observed, *Appendix 1—figure 2*. Compared to well-mixed results, *Appendix 1—figure 2a*, by decreasing the activation rate of CaMAC from 59.5 to 0.6 s$^{-1}$, a switch to out-of-phase can occur, *Appendix 1—figure 2c*. Yet, we notice how increasing basal PDE activity does not switch the phase *Appendix 1—figure 2b*, only after subsequently decreasing PDE and CaM association does the phase change *Appendix 1—figure 2d*. This is due to both changes being required for the activity variation of PDE to outperform that of AC. These findings lead to more questions about how source and sink activities relate in a variable $Ca^{2+}$ regime.

## Well-mixed reaction tables

**Appendix 1—table 1.** Voltage gated channel reactions.

| # | Species | Expression | Parameters | Ref. |
|---|---------|------------|------------|------|
| 1 | Membrane Voltage | $\frac{dV}{dt} = \frac{-I_{Ca}-I_K-I_L-I_{KCa}}{C_m}$ | $C_m$ = 5.3 pF | *Ni et al., 2011* |
| 2 | $Ca^{2+}$ current | $I_{Ca} = g_{Ca}m_\infty(V - E_{Ca})$ | $g_{Ca}$ = 600 pS, $E_{Ca}$ = 100 mV | *Ni et al., 2011* |
| 3 | Fraction of open VGCC (at steady state) | $m_\infty = \frac{1}{2}\left(1 + tanh\left(\frac{V-v_1}{v_2}\right)\right)$ | $v_1$ = -20 mV, $v_2$ = 24 mV | *Ni et al., 2011* |
| 4 | $K^+$ current | $I_K = g_K w(V - E_K)$ | $g_K$ = 240 pS, $E_{Ca}$ = -75 mV | *Ni et al., 2011* |
| 5 | Fraction of open $K^+$ channels (at steady state) | $\frac{dw}{dt} = \frac{\Phi(w_\infty - w)}{\tau}$ | $\phi$ = 35 $\frac{1}{s}$ | *Ni et al., 2011* |
| 6 | Time constant for $K^+$ channel open probability | $\tau = \frac{1}{cosh\left(\frac{V-v_3}{2v_4}\right)}$ | $v_3$ = -16 mV, $v_4$ = 11.2 mV | *Ni et al., 2011* |
| 7 | leak current | $I_L = g_L(V - E_L)$ | $g_L$ = 150 pS, $E_L$ = -75 mV | *Zhang et al., 2005* |
| 8 | $Ca^{2+}$ gated $K^+$ current | $I_{KCa} = g_{KCa}\frac{Ca}{Ca+K_{KCa}}(V - E_K)$ | $g_{KCa}$ = 2000 pS, $E_{Ca}$ = -75 mV, $K_{KCa}$ = 5 μ$M$ | *Ni et al., 2011* |

**Appendix 1—table 2.** $Ca^{2+}$ flux and reactions.

| # | Reaction | Reaction flux | Kinetic parameters | Ref. |
|---|----------|---------------|--------------------|------|
| 9 | → Ca | $j_{Ca_v}(1 + k_{PKA_v}[PKA]) + j_{Ca_l}$ | $k_{PKAV}$ = 3000 s$^{-1}$ · μM$^{-1}$ | *Ni et al., 2011* |
| 10 | $j_{Ca_v}$ | $f_i(-\alpha I_{Ca} - v_{LPM}[Ca])$ | $f_i$ = 1 x $10^{-5}$ $\alpha$ = 0.0045 μ$M$ · f$^{A-1}$ · s$^{-1}$ $v_{LPM}$ = 75 s$^{-1}$ | *Ni et al., 2011* |

*Continued on next page*

*Appendix 1—table 2 continued*

| # | Reaction | Reaction flux | Kinetic parameters | Ref. |
|---|---|---|---|---|
| 11 | $j_{Ca_I}$ | $\frac{Ck_{IP3R}[PKA]A[Ca]}{1+A[Ca]+[B][Ca]^2}([Ca_{stores}]-[Ca])-\frac{V_s[Ca]^2}{K_s^2+[Ca]^2}$ | $K_s$ = 10 μM, $Ca_{stores}$ = 1.56 μM, $V_s$ = 0.1 μM · s$^{-1}$, $k_{IP3R}$ = 0.05, A = 0.2869μM$^{-1}$, B = 2.869μM$^{-2}$, C = 0.2133 | *Ni et al., 2011* |
| 12 | 2Ca + CaM ↔ Ca$^2$CaM | $k_f[Ca][CaM]-k_r[Ca^2CaM]$ | $k_f$ = 3.6 s$^{-1}$ · μM$^{-1}$, $k_r$ = 8 s$^{-1}$ | *Lai et al., 2015* |
| 13 | Ca + Ca$^2$CaM ↔ Ca$^3$CaM | $k_f[Ca][Ca^2CaM]-k_r[Ca^3CaM]$ | $k_f$=11 s$^{-1}$ · μM$^{-1}$, $k_r$ = 195 s$^{-1}$ | *Lai et al., 2015* |
| 14 | Ca + Ca$^3$CaM ↔ Ca$^4$CaM | $k_f[Ca][Ca^3CaM]-k_r[Ca^4CaM]$ | $k_f$ = 59 s$^{-1}$·μM$^{-1}$, $k_r$ = 500 s$^{-1}$ | *Lai et al., 2015* |
| 15 | AC + Ca$^2$CaMCaM ↔ CaM · AC | $k_f[AC][Ca^2CaM]-k_r[CaM \cdot AC]$ | $k_f$ = 1.7 s$^{-1}$ · μM$^{-1}$, $k_r$ = 10 s$^{-1}$ | *Masada et al., 2009; Masada et al., 2012* |
| 16 | CaM · AC + 2Ca ↔ AC* | $K_{cat}\frac{[Ca][CaM \cdot AC]}{Ca+K_m}-k_r[AC *]$ | $K_{cat}$ = 59.5 s$^{-1}$, $K_m$ = 0.1 μM, $k_r$ = 10 s$^{-1}$ | *Masada et al., 2009; Masada et al., 2012* |
| 17 | PDE + Ca$^2$CaM ↔ CaM · PDE | $k_f[PDE][Ca^2CaM]-k_r[CaM \cdot PDE]$ | $k_f$ = 435 s$^{-1}$ · μM$^{-1}$, $k_r$ = 1 s$^{-1}$ | *Ang and Antoni, 2002* |
| 18 | CaM · PDE + 2Ca ↔ PDE* | $K_{cat}\frac{[Ca][CaM \cdot PDE]}{Ca+K_m}-k_r[PDE *]$ | $K_{cat}$ = 1.81 s$^{-1}$, $K_m$ = 0.18 μM, $k_r$ = 1 s$^{-1}$ | *Ang and Antoni, 2002* |
| 19 | PDE + Ca$^4$CaM ↔ PDE* | $k_f[PDE][Ca^4CaM]-k_r[PDE *]$ | $k_f$ = 435 s$^{-1}$ · μM$^{-1}$, $k_r$ = 1 s$^{-1}$ | *Ang and Antoni, 2002* |

*denotes the activated form

**Appendix 1—table 3.** cAMP reactions.

| # | Reaction | Reaction flux | Kinetic parameters | Ref. |
|---|---|---|---|---|
| 20 | → cAMP | $k_{base}([CaM \cdot AC]+[AC]+[AC_{ind}])+k_{act}[AC\star]$ | $k_{base}$ = 0.1 s$^{-1}$, $k_{act}$ = 0.785 s$^{-1}$ | Leonid E. Fridlyand & Philipson, 2016; *Ni et al., 2011* |
| 21 | cAMP→ | $k_{base}[cAMP]\frac{[CaM \cdot PDE]+[PDE]}{[cAMP]+K_m}+k_{act}\frac{[cAMP][PDE *]}{[cAMP]+K_m}$ | $k_{base}$ = 0.2 s$^{-1}$, $K_m$ = 0.6 μM $k_{act}$ = 2.5 s$^{-1}$ | *Ni et al., 2011; Fridlyand and Philipson, 2016* |
| 22 | cAMP→ | $V_{ind}\frac{[cAMP]}{[cAMP]+K_m}$ | $V_{ind}$ = 2.5 μM · s$^{-1}$, $K_m$ = 1.4 μM | *Ni et al., 2011; Fridlyand and Philipson, 2016* |
| 23 | cAMP + R2 → R2$_b$ | $k_f[cAMP][R2]-k_r[R2_b]$ | $k_f$ = 1 s$^{-1}$ · μM$^{-1}$, $k_r$ = 0.00033 s$^{-1}$ | *Boras et al., 2014* |
| 24 | cAMP + R2$_b$ → R2$_{ba}$ | $k_f[cAMP][R2_b]-k_r[R2_{ba}]$ | $k_f$ = 1 s$^{-1}$ · μM$^{-1}$, $k_r$ = 0.00105 s$^{-1}$ | *Boras et al., 2014* |

*Continued on next page*

*Appendix 1—table 3 continued*

| # | Reaction | Reaction flux | Kinetic parameters | Ref. |
|---|---|---|---|---|
| 25 | cAMP + R2$_b$ → R2$_{bb}$ | $k_f[cAMP][R2_b] - k_r[R2_{bb}]$ | $k_f = 1\ s^{-1} \cdot \mu M^{-1}$, $k_r = 0.00132\ s^{-1}$ | *Boras et al., 2014* |
| 26 | cAMP + R2$_{ba}$ → R2$_{bba}$ | $k_f[cAMP][R2_{ba}] - k_r[R2_{bba}]$ | $k_f = 1\ s^{-1} \cdot \mu M^{-1}$, $k_r = 0.0013\ s^{-1}$ | *Boras et al., 2014* |
| 27 | cAMP + R2$_{bb}$ → R2$_{bba}$ | $k_f[cAMP][R2_{bb}] - k_r[R2_{bba}]$ | $k_f = 1\ s^{-1} \cdot \mu M^{-1}$, $k_r = 0.00103\ s^{-1}$ | *Boras et al., 2014* |
| 28 | cAMP + R2$_{bba}$ → R2$_{bbaa}$ | $k_f[cAMP][R2_{bba}] - k_r[R2_{bbaa}]$ | $k_f = 1\ s^{-1} \cdot \mu M^{-1}$, $k_r = 0.0114\ s^{-1}$ | *Boras et al., 2014* |
| 29 | PKA + R2 → R2C | $k_f[PKA][R2] - k_r[R2C]$ | $k_f = 1\ s^{-1} \cdot \mu M^{-1}$, $k_r = 1.26E-7\ s^{-1}$ | *Boras et al., 2014* |
| 30 | PKA + R2$_b$ → R2$_{bc}$C | $k_f[PKA][R2_b] - k_r[R2_bC]$ | $k_f = 1\ s^{-1} \cdot \mu M^{-1}$, $k_r = 2.52E-7\ s^{-1}$ | *Boras et al., 2014* |

**Appendix 1—table 4.** cAMP reactions (cont.).

| # | Reaction | Reaction flux | Kinetic parameters | Ref. |
|---|---|---|---|---|
| 31 | PKA + R2$_{ba}$ → R2$_{ba}$ C | $k_f[PKA][R2_{ba}] - k_r[R2_{ba}C]$ | $k_f = 1\ s^{-1} \cdot \mu M^{-1}$ $k_r = 3.4E-6\ s^{-1}$ | *Boras et al., 2014* |
| 32 | PKA + R2$_{bba}$ → R2$_{bba}$ C | $k_f[PKA][R2_{bba}] - k_r[R2_{bba}C]$ | $k_f = 1\ s^{-1} \cdot \mu M^{-1}$, $k_r = 0.000936\ s^{-1}$ | *Boras et al., 2014* |
| 33 | PKA + R2$_{bbaa}$ → R2$_{bbaa}$ C | $k_f[PKA][R2_{bbaa}] - k_r[R2_{bbaa}C]$ | $k_f = 1\ s^{-1} \cdot \mu M^{-1}$, $k_r = 0.645\ s^{-1}$ | *Boras et al., 2014* |
| 34 | cAMP + R2C → R2$_b$ C | $k_f[cAMP][R2C] - k_r[R2_bC]$ | $k_f = 1\ s^{-1} \cdot \mu M^{-1}$, $k_r = 0.000659\ s^{-1}$ | *Boras et al., 2014* |
| 35 | cAMP + R2$_b$ C → R2$_{ba}$ C | $k_f[cAMP][R2_bC] - k_r[R2_{ba}C]$ | $k_f = 1\ s^{-1} \cdot \mu M^{-1}$, $k_r = 0.0142\ s^{-1}$ | *Boras et al., 2014* |
| 36 | cAMP + R2$_{ba}$ C → R2$_{bbc}$ C | $k_f[cAMP][R2_{ba}C] - k_r[R2_{bba}C]$ | $k_f = 1\ s^{-1} \cdot \mu M^{-1}$, $k_r = 0.0142\ s^{-1}$ | *Boras et al., 2014* |
| 37 | cAMP + R2$_{bba}$ C → R2$_{bbaa}$ C | $k_f[cAMP][R2_{bba}C] - k_r[R2_{bbaa}C]$ | $k_f = 1\ s^{-1} \cdot \mu M^{-1}$, $k_r = 7.84\ s^{-1}$ | *Boras et al., 2014* |
| 38 | PKA + R2$_b$ C → R2$_b$ C$_2$ | $k_f[PKA][R2_bC] - k_r[R2_bC_2]$ | $k_f = 1\ s^{-1} \cdot \mu M^{-1}$, $k_r = 0.00324\ s^{-1}$ | *Boras et al., 2014* |
| 39 | PKA + R2C → R2C$_2$ | $k_f[PKA][R2C] - k_r[R2C_2]$ | $k_f = 1\ s^{-1} \cdot \mu M^{-1}$, $k_r = 2.81E-6\ s^{-1}$ | *Boras et al., 2014* |
| 40 | PKA + R2$_{ba}$ C → R2$_{ba}$ C$_2$ | $k_f[PKA][R2_{ba}C] - k_r[R2_{ba}C_2]$ | $k_f = 1\ s^{-1} \cdot \mu M^{-1}$, $k_r = 0.666\ s^{-1}$ | *Boras et al., 2014* |
| 41 | cAMP + R2C$_2$ → R2$_b$ C$_2$ | $k_f[cAMP][R2C_2] - k_r[R2_bC_2]$ | $k_f = 1\ s^{-1} \cdot \mu M^{-1}$, $k_r = 0.762\ s^{-1}$ | *Boras et al., 2014* |
| 42 | cAMP + R2$_b$ C$_2$ → R2$_{ba}$ C$_2$ | $k_f[cAMP][R2_bC_2] - k_r[R2_{ba}C_2]$ | $k_f = 1\ s^{-1} \cdot \mu M^{-1}$, $k_r = 2.91\ s^{-1}$ | *Boras et al., 2014* |

**Appendix 1—table 5.** Initial conditions.

| # | Species | Initial value | Ref. |
|---|---|---|---|
| IC1 | AC8 | 1 µM | |
| IC2 | $AC_{ind}$ | 1 µM | |
| IC3 | $Ca^{2+}$ | 1 µM | |
| IC4 | CaM | 10 µM | |
| IC5 | cAMP | 0.1 µM | |
| IC6 | PDE1 | 1 µM | |
| IC7 | PDE4 | 0.4 µM | |
| IC8 | R2C2 | 0.4 µM | |
| IC9 | V | −60 mV | |

## Minimal model to explore $Ca^{2+}$-cAMP phase behavior

Based on the well-mixed model, we propose a minimal circuit to understand the phase behavior of $Ca^{2+}$-cAMP. Consider the system in *Figure 2c*, where $Ca^{2+}$ is the stimulus which uses a pulse train to set influx times, AC is an activator, PDE1C is an inhibitor, and cAMP is the response element.

### Assumptions

The system is in a state such that the change in cAMP allows for further $Ca^{2+}$ influx in a semi-predictable manner, as represented by the pulse train. Therefore, this system is deemed to be stable if there exists a $Ca^{2+}$ value that gives a stable solution for cAMP. All constants must be positive to remain physically relevant. We assume that there exists a constant independent source and sink within the system. The $Ca^{2+}$-dependent and independent sources are localized homogeneously on the membrane, with both sinks located uniformly in the cytosol. For simplicity, we assume that the activation function for both $AC$ and $PDE1C$ are linear functions of $Ca^{2+}$ of the form $aS + b$.

### Governing Equations

We define a stimulus (S, $Ca^{2+}$) and a response element (R, cAMP). The well-mixed rate of change function for cAMP is then given by,

$$\frac{dR}{dt} = v_1(a_1 S + b_1) - v_2(a_2 S + b_2)R + v_{ip} - v_{id}R$$

Here, $v_1$ denotes the velocity of cAMP production by AC, $v_2$ the rate of degradation by PDE, and $v_{ip}$ and $v_{id}$ the rates of independent production and degradation respectively.

### Analytical solutions for the minimal model

To analyze if the system lies in an in- or out-of-phase state we find the direction of the system change after initialization to $S_0$ [i.e. the basal stimulus (initial concentration of $Ca^{2+}$)]. First, we must solve for $R_0$ (initial concentration of cAMP) by setting $\frac{dR}{dt} = 0$, we find:

$$R_0 = \frac{v_1(a_1 S_0 + b_1) + v_{ip}}{v_2(a_2 S_0 + b_2) + v_{id}}$$

We start the system at equilibrium and prescribe a discontinuous pulse of $S$ from $S_0$ to $S_h$, akin to VGCC opening allowing $Ca^{2+}$ flux. Therefore, solving for the sign of R, we obtain

$$\frac{dR}{dt}\Big|_{t=0} = v_1(a_1 S_h + b_1) + v_{ip} - (v_2(a_2 S_h + b_2) + v_{id})\frac{v_1(a_1 S_0 + b_1) + v_{ip}}{v_2(a_2 S_0 + b_2) + v_{id}}$$

$$\frac{dR}{dt}\Big|_{t=0} = \frac{(v_1 a_1(v_2 b_2 + v_{id}) - v_2 a_2(v_1 b_1 + v_{ip}))(S_h - S_0)}{v_2(a_2 S_0 + b_2) + v_{id}}$$

Since we consider the sign we can then characterize the solution by

$$\frac{v_1 a_1(v_2 b_2 + v_{id})}{v_2 a_2(v_1 b_1 + v_{ip})} > 1 \qquad \text{inphase(low } \tau)$$

$$\frac{v_1 a_1(v_2 b_2 + v_{id})}{v_2 a_2(v_1 b_1 + v_{ip})} = 1 \qquad \text{transition(flatline)}$$

$$\frac{v_1 a_1(v_2 b_2 + v_{id})}{v_2 a_2(v_1 b_1 + v_{ip})} < 1 \qquad \text{outofphase(high } \tau)$$

This model was then computationally run to confirm the results with arbitrary parameters (*Figure 2c*). The system was simulated with all parameters set to 1 except $v_1$ and $v_2$ at values of 4 and 2, respectively and vice-versa. The pulse train is ON for 1 sec ($S_h$ = 10) and OFF for 3 sec ($S_0$ = 1), and repeats until the simulation total time is reached.

## Numerical implementation of the minimal model

We performed numerical simulations in MATLAB R2018b and checked against analytic solutions provided in the previous section. Minimal model solutions are shown in *Figure 2c*.

## Stochastic implementation of the minimal model

To further explore the possible effects of noise and small numbers of molecules on the simplified four component model, we created a stochastic version of the model. Due to the nature of stochastic simulations, all reactions were rewritten to multistep interactions. In order to allow for oscillatory kinetics at the membrane, a Lotka-Volterra model (in a larger volume, for numerical stability) was utilized to simulate depolarization events shown in *Appendix 1—table 8*. The system was solved using the Gibson method in the Virtual Cell modeling platform. All $Ca^{2+}$, AC, and PDE reactions are shown in *Appendix 1—tables 6–9*.

**Appendix 1—table 6.** Initial conditions for the stochastic model.

| Species | Number of molecules in the system |
|---|---|
| $Ca^{2+}$ | 10 |
| PDE | 45 |
| PDE* | 5 |
| AC | 10–200 (log scale) |
| s0 | 10000 |
| s4 | 1000 |

**Appendix 1—table 7.** Compartment volumes for the stochastic model.

| Compartment | Compartment size |
|---|---|
| Cytosol | 0.083 μm$^3$ |
| Membrane | 0.28 μm$^2$ |
| Oscillator | 6563.0 μm$^2$ |

**Appendix 1—table 8.** Lotka-Volterra model to initiate the membrane depolarization events.

| # | Reaction | Parameters |
|---|---|---|
| 1 | s0 → 2s0 | $K_f = 1.7\ s^{-1}$ |
| 2 | s0 + s4 → 2s4 | $K_f = 1\ \mu m^2 \cdot molecule^{-1} \cdot s^{-1}$ |
| 3 | s4 → | $K_f = 1.7\ s^{-1}$ |

**Appendix 1—table 9.** simplified model reactions for the stochastic model.

| # | Reaction | Parameters |
|---|---|---|
| 4 | s4 → > s4 + Ca$^{2+}$ | $K_f = 150\ s^{-1}$ |
| 5 | Ca$^{2+}$→ | $K_f = 500\ \mu m^{-2} \cdot molecule \cdot s^{-1} \cdot \mu M^{-1}$ |
| 6 | AC + Ca$^{2+}$ → AC* | $K_f = 200\ s^{-1} \cdot \mu M^{-1}$  $K_r = 300\ s^{-1}$ |
| 7 | AC* → > cAMP + AC* | $K_f = 300\ s^{-1}$ |
| 8 | AC → cAMP + AC | $K_f = 5\ s^{-1}$ |
| 9 | PDE + Ca$^{2+}$ → PDE* | $K_f = 150\ s^{-1} \cdot \mu M^{-1}$  $K_f = 100\ s^{-1}$ |
| 10 | PDE + cAMP → PDE | $K_f = 50\ s^{-1} \cdot \mu M^{-1}$ |
| 11 | PDE* + cAMP → PDE* | $K_f = 1000\ s^{-1} \cdot \mu M^{-1}$ |
| 12 | → cAMP | $J = 9000\ \mu m^{-2} \cdot molecule \cdot s^{-1}$ |
| 13 | cAMP → | $K_f = 50\ s^{-1}$ |

Solutions to the stochastic model are shown in *Figure 6—figure supplement 2*. The Virtual Cell Model, 'Minimum Stochastic Model BetaCell' by user 'mgetz', can be accessed within the VCell software (available at https://vcell.org). The Virtual Cell is supported by NIH Grant R24 GM134211 from the National Institute for General Medical Sciences (*Cowan et al., 2012*; *Schaff et al., 1997*).

## Simulations of the full spatial systems

For computational simplification, simulations were performed with a Gaussian profile on the top boundary, the size of the domain and Gaussian profile were informed by STORM images, *Figure 3*. The system is a hexagonal prism with outer diameter of 0.4 μm and depth 0.6 μm with periodic boundary conditions in the x and y planes, see *Figure 4a*. The top plane is assumed to be the membrane and the bottom is a no-flux condition. For the membrane plane, a Gaussian profile was normalized such that the average value is 4 x $10^{-10}$ mol/m$^2$. The Ca$^{2+}$ sensitive AC initial condition (Gaussian profile) is fixed for all simulation times by setting the diffusion constant to ≈ 0. To test the effect of localization, a percentage of AC mass was moved from the Gaussian profile into a uniform profile (a 25% Localization means that 25% of the total AC mass is within the Gaussian profile and the remaining 75% is within the uniform profile). Examples of surface profiles at multiple % localizations can be seen in *Figure 4b*.

## Assumptions

- Membrane patterns are pre-existing and not affected by a single signaling event such that no diffusion occurs.
- Clustering events were approximated by a Gaussian profile in the center of the hexagonal prism.

## Model development

Although the well-mixed system shows the ability to oscillate in and out-of-phase in a well-behaved manner, the spatial model will not produce this effect if a homogeneous boundary is present. Previous studies have looked at spatial gradients in the context of cAMP and PKA and found out that localization of some species is necessary for the system to demonstrate different behaviors (*Yang et al., 2016*). Therefore, when moving to a 3D spatial map, we must consider how the two solution regimes can be recovered. Experimental data suggest that AKAP dimerizes and may form oligomers (*Gao et al., 2011*; *Gold et al., 2011*), which is important for the function of these cells (*Zhang et al., 2013*). This could allow spatial instabilities like those seen in *Haselwandter al., 2015* used to describe post synaptic domains. The final Gaussian profile on the top boundary had the size of the domain and Gaussian profile informed by STORM images of AC clustering (*Figure 3*). Statistics of the images show that, on average, 90% of the AC sits within 54 nm of the cluster center. The system was determined to be 0.35 x 0.35 x 0.6 μm hexagonal prism with a Gaussian standard deviation of 25 nm.

The kinetic parameters used were the same as for the well-mixed model, except for a few cases in which tuning through surface/volume relationships was needed. Post fitting was performed to further refine the relationship between PDE and AC through use of obtained FRET measurements (see section 4). Due to the large computational expense of the model, the PDE interactions were reduced to two steps (*Appendix 1—table 10*).

**Appendix 1—table 10.** $Ca^{2+}$ Flux and reactions modified from *Appendix 1—table 2*.

| # | Reaction | Reaction flux | Kinetic parameters | Ref. |
|---|---|---|---|---|
| S1 | $\to Ca^{2+}$ | $j_{Ca_V}(1 + k_{PKA_V}[PKA]) + J_{Ca_I}$ | $k_{PKAV} = 100\ \mu M^{-1}$ | FRET constraint |
| S2 | $j_{Ca_V}$ | $f_i(-\alpha I_{Ca} - v_{LPM}[Ca])$ | $f_i = 1 \times 10^{-6}$, $\alpha = 4.15 \times 10^5 mol \cdot m^{-2} \cdot A^{-2} \cdot s^{-1}$,, $v_{LPM} = 7.5 \times 10^{-4} m \cdot s^{-1}$ | *Ni et al., 2011* |
| S3 | $j_{Ca_I}$ | $\frac{Ck_{IPAR}[PKA]A[Ca]}{1+A[Ca]+[B][Ca]^2}([Ca_{stores}] - [Ca]) - \frac{V_s[Ca]^2}{K_s^2+[Ca]^2}$ | $K_s = 10\ \mu M$, $Ca_{stores} = 1.56\ \mu M$, $V_s = 0.1\ \mu M \cdot s^{-1}$, $k_{IP3R} = 0.05\ \mu M^{-1}s^{-1}$, $A = 0.2869\ \mu M^{-1}$, $B = 2.869\ \mu M^{-2}$, $C = 0.2133$ | *Ni et al., 2011* |
| S4 | $AC + Ca^2CaM \leftrightarrow CaM \cdot AC$ | $k_f[AC][Ca^2CaM] - k_r[CaM \cdot AC]$ | $k_f = 10.8\ s^{-1} \cdot \mu M^{-1}$, $k_r = 10\ s^{-1}$ | FRET constraint |
| S5 | $CaM \cdot AC + 2Ca2+ \leftrightarrow AC*$ | $K_{cat}\frac{[Ca][CaM \cdot AC]}{Ca+K_m} - k_r[AC\ *]$ | $K_{cat} = 90\ s^{-1}$, $K_m = 1\ \mu M$, $k_r = 10\ s^{-1}$ | FRET constraint |
| S6 | $PDE + Ca^2CaM \leftrightarrow CaM \cdot PDE$ | $k_f[PDE][Ca^2CaM] - k_r[CaM \cdot PDE]$ | $k_f = 0.25\ s^{-1} \cdot \mu M^{-1}$, $k_r = 1\ s^{-1}$ | FRET constraint |
| S7 | $CaM \cdot PDE + 2Ca2+ \leftrightarrow PDE*$ | $K_{cat}\frac{[Ca][CaM \cdot PDE]}{Ca+K_m} - k_r[PDE\ *]$ | $K_{cat} = 60\ s^{-1}$, $K_m = 1\ \mu M$, $k_r = 1\ s^{-1}$ | FRET constraint |
| S8 | $PDE + Ca^4CaM \leftrightarrow PDE*$ | $k_f[PDE][Ca^4CaM] - k_r[PDE\ *]$ | $k_f = 0.25\ s^{-1} \cdot \mu M^{-1}$, $k_r = 1\ s^{-1}$ | FRET constraint |

## Numerical simulation

The well-mixed network was imported into COMSOL Multiphysics5.4 (Build:295), to solve the spatial model with inhomogeneous boundary conditions at the plasma membrane.

## Reaction-Diffusion Partial differential equations (PDEs)

Consider the same one-reaction system as the well-mixed system:

$$A + 2B \rightleftharpoons 3C,$$

where the forward and backward rates are $k_1$ and $k_2$. But, now we spatially discretize the system, the partial differential equations describing the dynamics of species $A, B$, and $C$ with unrestricted diffusion are then:

$$\frac{\partial[A]}{\partial t} = D_A \nabla^2[A] + k_2[C]^3 - k_1[A][B]^2$$
$$\frac{\partial[B]}{\partial t} = D_B \nabla^2[B] + 2k_2[C]^3 - 2k_1[A][B]^2$$
$$\frac{\partial[C]}{\partial t} = D_C \nabla^2[C] + 3k_1[A][B]^2 - 3k_2[C]^3.$$

Where $\nabla^2$ is the Laplacian and $D_A$ is a diagonal matrix of diffusion coefficients for component A. Yet within the cell, all reactions do not occur in the free volume. In fact, most interactions occur on a membrane surface, which requires a different boundary condition.

## Reactions on the system boundary

Now let us assume the previous reaction occurs on the boundary and C is a membrane species on surface $\Omega$

$$A + 2B \rightleftharpoons 3C,$$

This would mean within the volume there is only free diffusion,

$$\frac{\partial[A]}{\partial t} = D_A \nabla^2[A]$$
$$\frac{\partial[B]}{\partial t} = D_B \nabla^2[B]$$

At the surface we would have a Neumann boundary condition, that is, a defined flux occurring into the boundary normal. We define the reaction as a flux occurring at a membrane surface ($\Omega$)

$$\nabla A(\mathrm{x},t) \cdot \hat{n}(\mathrm{x})|_{\partial\Omega} = k_2[C]^3 - k_1[A][B]^2$$
$$\nabla B(\mathrm{x},t) \cdot \hat{n}(\mathrm{x})|_{\partial\Omega} = 2k_2[C]^3 - 2k_1[A][B]^2$$
$$\nabla C(\mathrm{x},t) \cdot \hat{n}(\mathrm{x})|_{\partial\Omega} = 3k_1[A][B]^2 - 3k_2[C]^3$$

Here, $\hat{n}$ is the unit normal to $\Omega$. We therefore have defined a flux between the volume and the membrane $\Omega$. Finally, we consider $\Omega$ as a 2-dimensional surface existing at the system boundary with free diffusion of C;

$$\frac{\partial[C]}{\partial t} = D_C \Delta[C]$$

where $\Delta$ is the surface Laplacian.

## PKA spatial response mirrors cAMP

PKA with full diffusion values ($\approx 10 \frac{\mu m^2}{s}$) did not follow cAMP dynamics and only elicited a single global response. We asked if this was due to AKAP patterning at the surface, which should mirror the AC profile at the surface. Adding this interaction into the model did not allow any sizable spatial gradient of PKA to develop. Recent work has suggested the PKA catalytic subunit in the presence of non-excess cAMP is effectively activated but its diffusion is restricted (*Smith et al., 2017*). By varying the diffusion constant, we found PKA activity could follow cAMP dynamics in the nanocluster and PM compartments in our computational model for restricted diffusivities (*Figure 6—figure supplement 1a*). We experimentally tested this prediction using the AKAP79-fused and PM-targeted PKA activity sensors and found that indeed PKA activity did follow the cAMP-Ca$^{2+}$ phase relationship within the two compartments (*Figure 6—figure supplement 1b-d*). This suggests that anchored PKA holoenzyme action is much more restricted than originally anticipated.

## Comparisons with experimental data

Raw FRET data (*Figure 1*) was used for model refinement of the cAMP-Ca$^{2+}$ phase relationship. The experimental and model data were qualitatively compared against each other to preserve oscillation time, amplitude, range, and waveform. Although a quantitative calibration was not performed to map FRET ratio to cAMP concentration, we utilized the fact that these sensors are known to measure cAMP in the range of $\approx$ 0.1-10 µM to constrain our model. Voltage-gated channel sensitivities were not tuned, and only connection strengths between CaM to ACs and PDEs, which are largely less constrained in comparison, were varied. Values modified from the well-mixed model values can be found in *Appendix 1—tables 10–13*.

## Model validation and predictions

The model generated predictions relating concentration perturbations (AC, PDE, etc.), disruption of patterning (AC binding disruption), and their changes to the phase of the signal. The system, once moved to the spatial model, was allowed free parameters along the six-component axis for CaM connection and cAMP production strength of ACs and PDEs. This includes the flux differential between basal and activated ACs and PDEs.

## Reaction tables of modified well-mixed parameters for the spatial model

**Appendix 1—table 11.** cAMP reactions modified from *Appendix 1—table 3*.

| # | Reaction | Reaction flux | Kinetic parameters | Ref. |
|---|---|---|---|---|
| S9 | → cAMP | $k_{base}([CaM \cdot AC] + [AC] + [AC_{ind}]) + k_{act}[AC *]$ | $k_{base}$ = 0.2 $s^{-1}$, $k_{act}$ = 23.55 $s^{-1}$, ACind = 3 x 10$^{-8}$ | FRET constraint |
| S10 | cAMP → | $k_{base}[cAMP]\frac{[CaM \cdot PDE]+[PDE]}{[cAMP]+K_m} + k_{act}\frac{[cAMP][PDE *]}{[cAMP]+K_m}$ | $k_{base}$ = 0.6 $s^{-1}$, $K_m$ = 0.6 µM $k_{act}$ = 720 $s^{-1}$ | FRET constraint |
| S11 | cAMP → | $V_{ind}\frac{[cAMP]}{[cAMP]+K_m}$ | $V_{ind}$ = 0.25 µM · $s^{-1}$, $K_m$ = 1.4 µM | FRET constraint |
| S12 | 2cAMP + R2C2 → R2C + PKA | $k_f[cAMP]^2[R2C2] - k_r[R2C][PKA]$ | $k_f$ = 20 $min^{-1}$ · µM$^{-2}$, $k_r$ = 12 $min^{-1}$ · µM$^{-1}$ | |
| S13 | 2cAMP + R2C → R2 + PKA | $k_f[cAMP]^2[R2C] - k_r[R2][PKA]$ | $k_f$ = 20 $min^{-1}$ · µM$^{-2}$, $k_r$ = 12 $min^{-1}$ · µM$^{-1}$ | |

**Appendix 1—table 12.** Additional AKAP interactions for the spatial model.

| # | Reaction | Reaction flux | Kinetic parameters | Ref |
|---|---|---|---|---|
| S14 | AKAP + R2 → AKAP-R2 | $k_f[R2][AKAP - R2C2] - k_r[AKAP - R2]$ | $k_f$ = 1 $s^{-1}$ · µM$^{-1}$ $k_f$ = 0.1 $s^{-1}$ | Est. |
| S15 | AKAP + R2C → AKAP-R2C | $k_f[R2C][AKAP - R2C2] - k_r[AKAP - R2C]$ | $k_f$ = 1 $s^{-1}$ · µM$^{-1}$ $k_f$ = 0.1 $s^{-1}$ | Est. |
| S16 | AKAP + R2C2 → AKAP-R2C2 | $k_f[R2C2][AKAP - R2C2] - k_r[AKAP - R2C2]$ | $k_f$ = 1 $s^{-1}$ · µM$^{-1}$ $k_f$ = 0.1 $s^{-1}$ | Est. |
| S17 | 2cAMP + AKAP-R2C2 → AKAP-R2C + PKA | $k_f[cAMP]^2[AKAP - R2C2] - k_r[AKAP - R2C][PKA]$ | $k_f$ = 20 $min^{-1}$ · µM$^{-2}$, $k_r$ = 12 $min^{-1}$ · µM$^{-2}$ | |

*Continued on next page*

*Appendix 1—table 12 continued*

| # | Reaction | Reaction flux | Kinetic parameters | Ref |
|---|---|---|---|---|
| S18 | 2cAMP + AKAP-R2C → AKAP-R2 + PKA | $k_f[cAMP]^2[AKAP-R2C] - k_r[AKAP-R2][PKA]$ | $k_f = 20\ min^{-1} \cdot \mu M^{-2}$, $k_r = 12\ min^{-1} \cdot \mu M^{-2}$ | |

**Appendix 1—table 13.** Initial conditions and diffusion coefficients used in the model.

| # | Species | Initial value | Diffusion | Ref. |
|---|---|---|---|---|
| SIC1 | $Ca^{2+}$ | 0.001 μM | $100\ \frac{\mu m^2}{s}$ | (**Donahue and Abercrombie, 1987**), Est. from steady state |
| SIC2 | CaM | 2.9 μM | $10\ \frac{\mu m^2}{s}$ | Est. from steady state* |
| SIC3 | $Ca_2CaM$ | 0.1 μM | $10\ \frac{\mu m^2}{s}$ | Est. from steady state* |
| SIC4 | $Ca_3CaM$ | $4 \times 10^{-3}$ μM | $10\ \frac{\mu m^2}{s}$ | Est. from steady state* |
| SIC5 | $Ca_4CaM$ | $1 \times 10^{-2}$ μM | $10\ \frac{\mu m^2}{s}$ | Est. from steady state* |
| SIC6 | $R_2$ | 0.04 μM | $10\ \frac{\mu m^2}{s}$ | Est. from steady state* |
| SIC7 | $R_2C_2$ | 0.2 μM | $10\ \frac{\mu m^2}{s}$ | Est. from steady state* |
| SIC8 | PKA | 0.05 μM | $0.01\ \frac{\mu m^2}{s}$ | Est. from steady state, diffusion fitted |
| SIC9 | PDE1 | 0.9 μM | $10\ \frac{\mu m^2}{s}$ | Est. from steady state* |
| SIC10 | $PDE1_{act}$ | $1 \times 10^{-3}$ μM | $10\ \frac{\mu m^2}{s}$ | Est. from steady state* |
| SIC11 | CaMPDE1 | $1 \times 10^{-3}$ μM | $10\ \frac{\mu m^2}{s}$ | Est. from steady state* |
| SIC12 | AC | $4 \times 10^{-10}\ \frac{mol}{m^2}$ | 0 | Est. |
| SIC13 | CaMAC | $0\ \frac{mol}{m^2}$ | 0 | Est. |
| SIC14 | $AC_{act}$ | $0\ \frac{mol}{m^2}$ | 0 | Est. |
| SIC15 | $AC_{ind}$ | $4 \times 10^{-10}\ \frac{mol}{m^2}$ | $1\ \frac{\mu m^2}{s}$ | Est.[†] |
| SIC16 | V | −60 mV | $1\ \frac{\mu m^2}{s}$ | **Fridlyand et al., 2003**; **Ni et al., 2011**[†] |
| SIC17 | cAMP | $4 \times 10^{-6}$ μM | $60\ \frac{\mu m^2}{s}$ | **Agarwal et al., 2016**, Est. from steady state |

*For all cytosolic species without well-constrained diffusions, we used $10\ \frac{\mu m^2}{s}$

[†]For all membrane species without well-constrained diffusions we used $1\ \frac{\mu m^2}{s}$

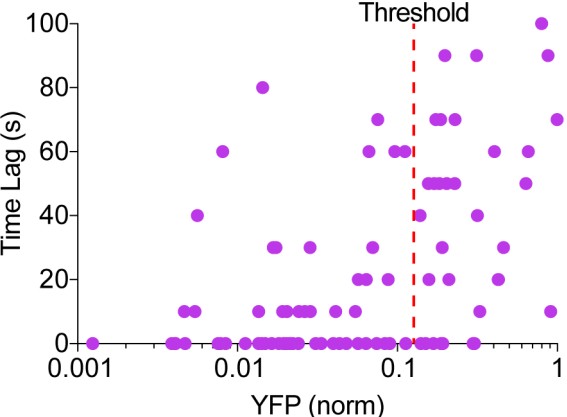

**Appendix 1—figure 1.** Phase of cAMP correlates with the expression level of the AKAP79-(Ci/Ce) Epac2-camps. Scatter plot of the time lag (sec) and the YFP donor channel intensity (normalized to non-saturating maximum) for each cell expressing AKAP79-(Ci/Ce)Epac2-camps. Cells with higher expression of the probe correlated with a longer time lag, therefore a YFP intensity threshold was designated for analysis purposes.

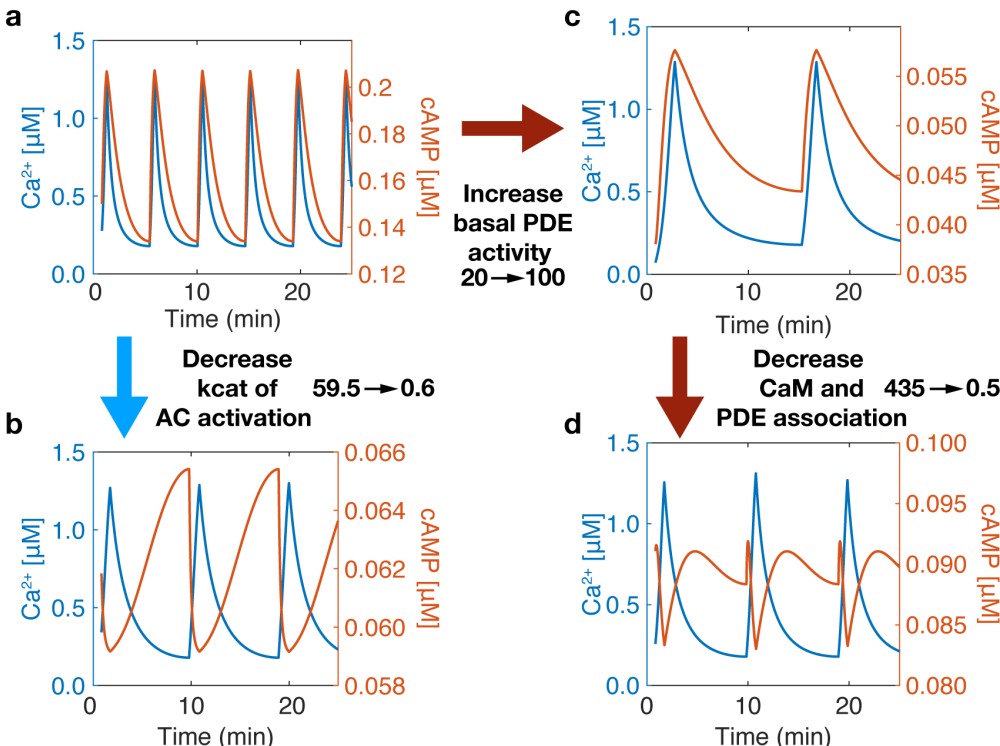

**Appendix 1—figure 2.** Phase is driven by activity variability within the $Ca^{2+}$ oscillatory regime. The phase of the system can be switched by tuning the association of CaM to sources (ACs) and sinks (PDEs). (**a**) At base system conditions, the system acts in an in-phase manner. (**b**) Decreasing the rate of $Ca^{2+}$ association to the AC-CaM complex causes the phase to switch to out-of-phase. (**c**) Increasing basal PDE activity does not allow a phase switch only after decreasing PDE and CaM association rates will the system allow a phase switch (**d**). A phase switch is controlled by the variability in the activity of source or sink. If the sink dominates, then the system is out-of-phase. If the source dominates, the system is in-phase.

