## [Decision Letter]

**Acceptance summary:**

Second messengers, typically small molecules such as cAMP and ions such as Ca^2+^, regulate a variety of intracellular processes. Understanding their spatial and temporal distributions as well as their dynamics is important to appreciate the spatial biochemistry controlled by these agents. In their manuscript the authors have been able to map exactly this for the oscillations of cAMP and Ca^2+^. Their analysis reports that the local oscillations of cAMP occur out-of-phase with respect to the oscillations of Ca^2+^, but in-phase in the rest of the cell. This has implications for the spatial and temporal control of the biochemistry regulated by this second messenger.

**Decision letter after peer review:**

Thank you for submitting your article "Spatially compartmentalized phase regulation of a Ca^2+^-cAMP-PKA oscillatory circuit" for consideration by *eLife*. Your article has been reviewed by three peer reviewers, and the evaluation has been overseen by a Reviewing Editor and Jonathan Cooper as the Senior Editor. The following individual involved in review of your submission has agreed to reveal their identity: André Nadler (Reviewer #3).

The reviewers have discussed the reviews with one another and the Reviewing Editor has drafted this decision to help you prepare a revised submission.

Summary:

The manuscript by Tenner et al. describes how spatial confinement of AMP cyclases can entrain the temporal concentration profile of its product cAMP to an oscillating calcium concentration profile. They address an extremely interesting topic that is currently receiving well-deserved attention: The notion that cells encode information not only through on/off switches in the respective signaling circuits but also in the dynamic profiles of the signaling responses. They provide compelling evidence for a two-state system featuring either in-phase or out-of-phase cAMP/calcium oscillations. They use FRET sensors targeted either to a specific (signaling) nano-compartment (AKAP79/150) at the plasma membrane or to the general plasma membrane to demonstrate that these states are due to distinct signaling dynamics on the nanoscale, a conclusion that is further underlined by simulations from a kinetic model describing this behavior.

They propose that a cluster of multiple calcium-sensitive ACs in the membrane shift the balance for the calcium-sensitive cAMP pool to become in-phase with the calcium signal itself, at increased calcium levels. In contrast, in regions away from the clusters, calcium-sensitive cAMP degradation by PDEs during periods of increased calcium drives the cAMP levels to become out-of-phase. In fact, the study goes full circle in illustrating how the spatially induced driving of a phase shift (calcium -> cAMP) could be related to the maintenance of the calcium oscillatory signal itself (via PKA). In general, the authors provide a well written narrative that includes microscopy of protein organisation at the nanometer scale, local and dynamic measurements of activity, and a modelling frame to interpret the data. Overall, this is a very interesting manuscript which would be of considerable general interest (and suitable for *eLife*) after a number of revisions to strengthen the experimental support for its core hypothesis.

Essential revisions:

A) AC8 nanoclustering and measurement of localized cAMP:

1) Although the STORM images of AKAP79 and AC8 make a plausible case that they are colocalized in nanoclusters a more convincing case would be made if the authors had carried out 2-color STORM.

The authors suggest that a cluster needs to contain 'many' AC8 proteins. Please provide a number: The authors have STORM data as well as simulations. For the reaction diffusion model the authors need to assume that the membrane patterns (AKAP79/150 – AC8 clusters) are immobile. If the domain should be immobile not with respect to a calcium-cAMP oscillation (on the order of 10s of seconds) it would be important to provide data to support this (FRAP, SPT or sequential imaging). How does a cluster of a few molecules of AC8, along with its scaffolding protein remain immobile in the fluid membrane? Is it known what generates and holds the AKAP79/150 compartment in place?

2) How critical is AC8 nanoclustering? Clearly under endogenous expression levels a displacement from the AKAP79/150 compartment disrupts the regularity of the calcium oscillations. But that is under the condition of the PDE activity that is present everywhere in the cytosol. What if the requirement is a calcium-dependent increase of cAMP production above the baseline? Then simple over expression of AC8 should be enough to entrain the calcium oscillations properly. Indeed, Figure 2E seems to suggest that to be a possibility. How critical is nanoclustering of AC8 in such a scenario of overexpression?

B) Sensor design

While the utilized sensor design based on the human protein may be of broader general interest, here a human protein (AKAP79) is used for targeting to a signaling nanodomain in a mouse cell line. The human and the mouse proteins are only 53% identical, thus it is not clear if they can in fact functionally substitute for each other, ensuring that the sensor indeed localized to the AKAP150/79 compartment.

This could for instance be confirmed using the already utilized AKAP150 antibody and an anti-FP antibody in a STORM (or other super-resolution) colocalization experiment. It would be better to generate a construct using the mouse protein and show in representative datasets that the observed phase-shift of cAMP oscillations is still observed. It is also not clear how the presence of a second protein pool (with likely different affinities for interacting proteins), in addition to the endogenous AKAP150 pool, would affect the outcome of the modelling. Possible complications may be discussed.

C) Perturbation of protein concentrations

The authors use overexpression of proteins as perturbations (e.g. for AC8). They do not, however, provide data on the actual changes of protein levels – only the amount of plasmid used for transfection is reported in Figure 2—figure supplement 2. The authors should quantify the changes of AC8 expression on the protein level, ideally in absolute numbers, however, if this is not possible, a combination of immunofluorescence data (to give the reader an estimate of the cell-to-cell variability) and western blot data would be sufficient here.

Does AKAP79 overexpression have an effect on endogenous AC8 expression levels?

Given that AKAP150 and AC8 physically interact, it is possible that their expression levels are co-regulated. Thus, it is at least conceivable that AK79 overexpression (through the FRET construct) could change AC8 expression levels. Given that AC8 concentration is a key factor in determining the phase offset of calcium/cAMP oscillations, this should be tested by quantifying AC8 on the protein level for overexpression of both the Lyn- and the AKAP79 based FRET constructs.

This could be done by western blot or MS to get an idea on the effect on the cell population level, but also by immunofluorescence imaging data on the single cell level (colocalization experiments using both anti-FP and anti-AC8 antibodies to estimate the effects of cell-to-cell variability).

D) Modelling Issues:

The modelling is competently done with well-established software in the field. The authors have made a couple of design decisions that should be probed: (1) the use of Michaelis-Menten enzyme kinetics, rather than multi-step mass-action (2) Not using stochastic calculations. The stochastic analysis would have yielded some estimates of noise in the system that might be interesting. While the mapping between model and experiment seems reasonable, an estimate of noise in the system due to the low numbers of the molecules involved would be worth exploring.

The authors also present two models which both explain the experimental data: (a, in Figure 2) A simpler one, where the concentration of AC8 serves as a (global) switch between out-of-phase and in-phase calcium and cAMP oscillations and (b, in Figure 4) a more complex one that predicts localized cAMP oscillations in the specific AKAP79/150 clusters which are out-of-phase with the (global) cAMP oscillations in the cytosol. All modelling is restricted to simulations, if the statement made in the supplementary information (subsection “Model development”) is to be interpreted correctly. This is a significant limitation for the entire manuscript, as all conclusions are (at least partially) based on choices for parameters that the authors made instead of parameters values that were derived from fitting the model to experimental data which would be much more stringent.

Ideally, the authors should fit their two models to a large, representative dataset and use Akaike's information criterion to decide which one is the more suited one. I understand that this will likely not be possible due to the high number of free parameters – it is unlikely that a fitting algorithm would find a true global minimum, as the available data probably are not sufficient constrains. However, this in turn means that deciding which model is to be preferred has to be done by testing predictions where the two models differ significantly.

The following two predictions where both models would yield diametrically opposing outcomes:

i) The 3D reaction diffusion model predicts that cAMP oscillations in the AKAP79/150 compartments and cAMP oscillations in the bulk cytosol are out of phase in the same cell (see Figure 4). This can be tested by combining the AKAP79-based FRET sensor with a cytosolic, intensiometric, RFP-based cAMP sensor such as R-FlincA (Ohta et al., 2018). If the authors observe a clear phase shift between these two reporters, it would constitute very strong evidence in support of the 3D reaction-diffusion model. The current datasets only report one type of compartment per cell, yet the striking difference between compartments in the same cell is where the model predictions differ most dramatically.

ii) Localized oscillations depend on the assumption that AC diffusion is (close to) zero. Thus, the authors should measure AC diffusion at the plasma membrane. A key assumption of their model is that at least a fraction of the total protein pool should be completely immobile. Furthermore, this fraction should increase in an AKAP150 overexpression background.

E) Final punchline that these elegant biophysical findings are important for some aspect of physiology is not very compelling. The authors should come with a more surgically-dissected physiological parameter that depends on the phase relationship between Ca and CaM, and at least provide this in their Discussion section.

---

## [Author Response]

Essential revisions:A) AC8 nanoclustering and measurement of localized cAMP:1) Although the STORM images of AKAP79 and AC8 make a plausible case that they are colocalized in nanoclusters a more convincing case would be made if the authors had carried out 2-color STORM.

We thank the reviewer for this suggestion. We tested different protocols for visualizing AC8 and AKAP150 nanoclusters in the same fixed cell and finally successfully performed 2-color STORM experiments (Figure 3—figure supplement 1). We quantified the degree of co-clustering of AC8 and AKAP150 using Getis-Franklin analysis (Getis and Franklin, 1987; Mo et al., 2017). We found that approximately 72% of AKAP150 localizations are co-clustered with AC8 localizations (Figure 3—figure supplement 1). These data corroborate our earlier data showing AC8:AKAP150 interaction (Figure 5—figure supplement 1A,B), further strengthen the validity of our spatial model, and have been incorporated into the revised manuscript.

The authors suggest that a cluster needs to contain 'many' AC8 proteins. Please provide a number: The authors have STORM data as well as simulations.

We thank the reviewers for this comment. Estimating the number of AC8 molecules from simulations is not possible because the simulations use a continuum description of the chemical species with the goal of identifying possible mechanisms underlying the in-phase and out-of-phase dynamics of calcium and cAMP. For ease of computation, we use surface densities and concentrations of the various species involved. Since we do not use molecular models of different species interaction, we cannot use the simulations to estimate the number of AC8 molecules. That sort of estimate requires molecular dynamic simulations that consider the structure of AC8 and AC8 membrane interactions, which is outside the scope of the present work.

Experimentally, counting the absolute number of proteins within cellular structures with STORM/PALM has traditionally been very difficult due to blinking artifacts, unknown labeling efficiencies, and specific acquisition parameters (Durisic et al., 2014; Jung, Fujimoto and Chiu, 2017) and is a very active field of investigation (Zanacchi et al., 2017; Thevathasan et al., 2019). However, we can obtain a very rough estimate of AC8 molecules within nanoclusters from our blink-corrected STORM datasets by assuming background localizations outside the cell are due to single secondary antibodies with conjugated dyes. Estimating a labeling stoichiometry of 0.4-1.5 secondary antibodies per primary antibody and 0.4-1.5 primary antibodies per AC8 molecule (Zanacchi et al., 2017; Nieuwenhuizen et al., 2015), we can roughly approximate the number of AC8 molecules per nanocluster to be between 15 to 150 molecules/cluster (N=3 cells). We have included this estimation in the Discussion section of the revised manuscript.

For the reaction diffusion model the authors need to assume that the membrane patterns (AKAP79/150 – AC8 clusters) are immobile. If the domain should be immobile not with respect to a calcium-cAMP oscillation (on the order of 10s of seconds) it would be important to provide data to support this (FRAP, SPT or sequential imaging). How does a cluster of a few molecules of AC8, along with its scaffolding protein remain immobile in the fluid membrane? Is it known what generates and holds the AKAP79/150 compartment in place?

Indeed, our simulations assume that the AKAP79/150-AC8 clusters are relatively immobile or diffuse slowly, within the timescale of the oscillatory circuit. Per the reviewer’s request, we performed Fluorescence Recovery After Photobleaching (FRAP) experiments using GFP-tagged AC8 and found that AC8 indeed diffuses slowly with a t_1/2_ recovery of 61 ± 20 sec (avg ± SD) and an apparent diffusivity D_app_ of 0.019 ± 0.002 μm^2^/sec (Figure 3—figure supplement 2, n = 16 cells). In addition, recovery of AC8 is incomplete, indicating a large immobile fraction at the membrane (extrapolated, avg. immobile fraction = 42.3%). These new data are included as Figure 3—figure supplement 2 in the revised manuscript. Restricted diffusion and clustering of receptors and scaffolds at the membrane is observed in many biological systems and in some cases can be attributed to an association with cytoskeletal elements (Sanderson, 2012; Trimble and Grinstein, 2015). Relevant to this study, clustering of the AKAP79/150 scaffold has been observed in other contexts (Mo et al., 2017; Zhang et al., 2016) and is known to interact with actin and cadherins (Gorski et al., 2005). It will be interesting in the near future to further investigate the role played by the AKAP scaffold in the observed restricted diffusion of AC8.

2) How critical is AC8 nanoclustering? Clearly under endogenous expression levels a displacement from the AKAP79/150 compartment disrupts the regularity of the calcium oscillations. But that is under the condition of the PDE activity that is present everywhere in the cytosol. What if the requirement is a calcium-dependent increase of cAMP production above the baseline? Then simple over expression of AC8 should be enough to entrain the calcium oscillations properly. Indeed, Figure 2E seems to suggest that to be a possibility. How critical is nanoclustering of AC8 in such a scenario of overexpression?

We thank the reviewer for these questions. In this study, we provide evidence that not only does entrainment of calcium and calcium-dependent increases of cAMP occur within MIN6 β cells, but rather these in-phase oscillations occur in a compartmentalized fashion. Based on our modeling analysis, AC8 nanoclustering is necessary but not sufficient for compartmentalized “phase” regulation, i.e. in-phase calcium and cAMP oscillations only occurring within the AKAP79/150-specific compartment. As shown in Figure 5B, disruption of the AC8 nanoclustering leads to loss of in-phase oscillations. We also modeled the case where AC8 is overexpressed but nanoclustering still exists, which we now present in Figure 4—figure supplement 1. In this case, increasing the concentration of nanoclustered AC8 by 50% drives calcium-stimulated cAMP oscillations to be in-phase with calcium both in the AKAP79/150-AC8 cluster center as well as outside the cluster at the plasma membrane (Figure 4—figure supplement 1). This is due to the additional calciumactivated cAMP flux from the cluster counteracting the cAMP degradation by PDE1 at the cytosol and general plasma membrane, which is consistent with the in-phase cAMP-calcium relationship that we observed at the general plasma membrane when AC8 was overexpressed (Figure 2C,E). Thus AC8 overexpression disrupts the compartmentalization and leads to calcium and cAMP entrainment in a global manner (Figure 2C,E and Figure 4—figure supplement 1).

B) Sensor designWhile the utilized sensor design based on the human protein may be of broader general interest, here a human protein (AKAP79) is used for targeting to a signaling nanodomain in a mouse cell line. The human and the mouse proteins are only 53% identical, thus it is not clear if they can in fact functionally substitute for each other, ensuring that the sensor indeed localized to the AKAP150/79 compartment.This could for instance be confirmed using the already utilized AKAP150 antibody and an anti-FP antibody in a STORM (or other super-resolution) colocalization experiment. It would be better to generate a construct using the mouse protein and show in representative datasets that the observed phase-shift of cAMP oscillations is still observed. It is also not clear how the presence of a second protein pool (with likely different affinities for interacting proteins), in addition to the endogenous AKAP150 pool, would affect the outcome of the modelling. Possible complications may be discussed.

We thank the reviewer for highlighting the important distinction between AKAP79 and AKAP150. Human AKAP79 and the rodent ortholog AKAP150 indeed share 53% sequence homology, however, the interaction motifs and association between the AKAP scaffold and the key signaling players (i.e. PKA, voltage-gated calcium channel, adenylyl cyclases, calcineurin, etc.) are conserved (Willoughby et al., 2010; Zhang and Shapiro, 2016). The primary difference between the two closely related scaffolds is the presence of an internal repetitive amino acid sequence of unknown function in AKAP150 (Robertson et al., 2009). The functional equivalence of AKAP79 and AKAP150 is demonstrated in several reciprocal knock-out and recovery experiments in which AKAP79 or AKAP150 is knocked out and expression of the other recovers a measured phenotype (Hoshi, Langeberg and Scott, 2005; Zhang et al., 2011). Specifically, relevant to this study, AC8 has also been shown to interact with both AKAP79 and AKAP150 (Willoughby et al., 2010). Thus, we believe AKAP79 can be considered functionally equivalent to AKAP150 in our experimental context and can report the AKAP150 compartment-specific signaling dynamics in the MIN6 β cells. We have incorporated these additional discussions about AKAP79 and AKAP150 in the revised manuscript.

We apologize for the lack of clarity regarding how expression of the AKAP79 scaffold might affect endogenous AKAP150-specific signaling dynamics. Indeed, signaling scaffold expression is tightly regulated in many biological systems in order to ensure association and stoichiometry of key enzymes within macromolecular complexes (Levchenko, Bruck and Sternberg, 2000). Thus, the effects of AKAP79 expression must be considered in this compartmentalized signaling study. Interestingly, we found a correlation between the expression level of the AKAP79-tethered cAMP sensor (AKAP79-Epac2-camps) and the observed lag time between calcium and AKAP79-specific cAMP (Appendix—figure 1). Cells with high expression of the AKAP79 sensor correlated with longer lag times and closely resembled the out-of-phase cAMP-calcium relationship at the general plasma membrane (Figure 1D). This suggests that additional AKAP scaffolds might “dilute” the associated signaling proteins per scaffold including, but not limited to, AC8, and this stoichiometric change might have downstream consequences. This balance would likely be interesting to investigate in the future, both computationally and experimentally; however, for the present study, we chose an empirically-derived expression cutoff in order to study and compare AKAP79/150 specific signaling between cells (Appendix—figure 1).

C) Perturbation of protein concentrationsThe authors use overexpression of proteins as perturbations (e.g. for AC8). They do not, however, provide data on the actual changes of protein levels – only the amount of plasmid used for transfection is reported in Figure 2—figure supplement 2. The authors should quantify the changes of AC8 expression on the protein level, ideally in absolute numbers, however, if this is not possible, a combination of immunofluorescence data (to give the reader an estimate of the cell-to-cell variability) and western blot data would be sufficient here.

We thank the reviewers for this suggestion. Originally, we overexpressed untagged AC8 due to the limited spectral space we had at our disposal (red RCaMP and cyan-yellow PM-targeted Epac2-camps). In order to measure the relative expression of transfected AC8, we tagged AC8 with GFP and transfected 50ng, 250ng, and 1000ng in MIN6 β cells, as previously described. After thresholding for GFP+ cells, we observed a positive correlation in the average GFP intensity per cell for increasing amounts of GFP-AC8 plasmid, indicating that indeed we are titrating the AC8 concentration in the perturbation experiment (Figure 2—figure supplement 2B). We have incorporated these data into a panel in Figure 2—figure supplement 2.

Does AKAP79 overexpression have an effect on endogenous AC8 expression levels?Given that AKAP150 and AC8 physically interact, it is possible that their expression levels are co-regulated. Thus, it is at least conceivable that AK79 overexpression (through the FRET construct) could change AC8 expression levels. Given that AC8 concentration is a key factor in determining the phase offset of calcium/cAMP oscillations, this should be tested by quantifying AC8 on the protein level for overexpression of both the Lyn- and the AKAP79 based FRET constructs.This could be done by western blot or MS to get an idea on the effect on the cell population level, but also by immunofluorescence imaging data on the single cell level (colocalization experiments using both anti-FP and anti-AC8 antibodies to estimate the effects of cell-to-cell variability).

Per the reviewer’s request, we transfected MIN6 β cells with lyn-Epac2-camps or AKAP79-Epac2-camps and stained for endogenous AC8 in order to see if expression of the tagged constructs affected AC8 expression.

At the single cell level, we assayed endogenous AC8 expression using immunocytochemistry and we found no significant difference in the average AC8 intensity between cells expressing the targeted cAMP sensors and untransfected cells (Figure 2—figure supplement 3A).

To assess the potential expression effects on a population level, we performed Western blots and probed for AC8 in MIN6 cells transfected with the targeted cAMP sensors or Cerulean alone (as a transfection/expression control). We measured robust transfection and expression efficiency of the sensors as well as the Cerulean-only control by probing for fluorescent protein (Figure 2—figure supplement 3B). Although we measured a small increase in AC8 expression for the two targeted sensor samples compared to the Cerulean only control (AKAP79, P = 0.0134; lyn, P = 0.0979), we found no significant difference in AC8 expression between the two targeted sensors (P = 0.4373, Figure 2—figure supplement 3B, N = 4 replicates). This demonstrates that the compartmentalized cAMP-Ca^2+^ phase difference we observe is not due to different expression levels of endogenous AC8. These new data are included as Figure 2—figure supplement 3.

D) Modelling Issues:The modelling is competently done with well-established software in the field. The authors have made a couple of design decisions that should be probed: (1) the use of Michaelis-Menten enzyme kinetics, rather than multi-step mass-action (2) Not using stochastic calculations. The stochastic analysis would have yielded some estimates of noise in the system that might be interesting. While the mapping between model and experiment seems reasonable, an estimate of noise in the system due to the low numbers of the molecules involved would be worth exploring.

We thank the reviewer for raising these issues. The choice of using Michaelis-Menten enzyme kinetics as opposed to multi-step mass-action was made on the basis of limiting the number of free parameters in the reaction model while retaining the kinetic features. Multistep mass-action models are often used in cases where all the intermediary states are known and sufficient kinetic data exists to resolve the rates of the intermediary steps. In this case, since our focus was on calcium-cAMP dynamics, we chose to collapse the many reaction steps into a Michaelis-Menten framework.

To strengthen this assumption, we conducted simulations using multistep reaction kinetics and the Michaelis-Menten formulation. As shown in Author response image 1, there was no discernible difference in the kinetics of AC and PDE. In the multistep reaction formulation, we needed to add 3 parameters that were not present in the Michaelis-Menten formulation. Based on this analysis, we retained the Michaelis-Menten formulation. The multi-step model was solved with kinetic parameter values found in Table 1.

Table 1: Kinetic parameters for the multi-step model plotted in Author response image 1.

**Author response image 1. sa2fig1:** of CaM^+2^Ca, CaMAC^+2^Ca, and CaMPDE^+2^Ca, were represented as multi step reactions with rates in the table given below. The system still exhibits in-phase and out-of-phase qualitative responses, with an additional 3 parameters. Since additional parameters did not affect outcome, we retained the Michaelis-Menten formulation.

In order to probe the role of low molecule number in determining the dynamics of calcium and cAMP in the model, we have rewritten the entire model to be compatible with Virtual Cell’s stochastic solver, built on Smoldyn. We have thus far verified that the deterministic model in Virtual Cell gives us results similar to those obtained in the finite element modeling software COMSOL. This step is important to establish the model outcome to be independent of solver modalities (Virtual Cell uses finite volume methods while COMSOL uses finite element methods).

Next, we translated the four-component deterministic model into its stochastic counterpart. Most stochastic solvers require that the reactions be written in their elementary form; therefore, we rewrote the Michaelis-Menten kinetic reactions into multi-step mass-action kinetics.

We recast our network motif model comprised of Ca^2+^, cAMP, PDE1, and AC8 as a stochastic system of equations and ran multiple simulations to approximate the probabilistic response of cAMP to an oscillatory Ca^2+^ input signal (details are presented in Appendix 1—table 6, Appendix 1—table 7, Appendix 1—table 8, Appendix 1—table 9). We find that the in-and out-of-phase relationships observed in the deterministic model are also found in the stochastic model. These important findings suggest that the oscillatory phase relationship persists at the scale of individual molecules and supports the previous results of our deterministic modeling (Figure 6—figure supplement 2A-C). Furthermore, we found that the level of noise in the system is reduced as the number of AC8 molecules increases (Figure 6—figure supplement 2D). This conclusion is interesting and suggests that noise in cAMP production/degradation can be curtailed when AC8 counts are high due to clustering or other methods of co-association. These new data are included in the revised manuscript as Figure 6—figure supplement 2.

The authors also present two models which both explain the experimental data: (a, in Figure 2) A simpler one, where the concentration of AC8 serves as a (global) switch between out-of-phase and in-phase calcium and cAMP oscillations and (b, in Figure 4) a more complex one that predicts localized cAMP oscillations in the specific AKAP79/150 clusters which are out-of-phase with the (global) cAMP oscillations in the cytosol. All modelling is restricted to simulations, if the statement made in the supplementary information (subsection “Model development”) is to be interpreted correctly. This is a significant limitation for the entire manuscript, as all conclusions are (at least partially) based on choices for parameters that the authors made instead of parameters values that were derived from fitting the model to experimental data which would be much more stringent.Ideally, the authors should fit their two models to a large, representative dataset and use Akaike's information criterion to decide which one is the more suited one. I understand that this will likely not be possible due to the high number of free parameters – it is unlikely that a fitting algorithm would find a true global minimum, as the available data probably are not sufficient constrains. However, this in turn means that deciding which model is to be preferred has to be done by testing predictions where the two models differ significantly.

We apologize for any confusion and would like to clarify an important point here – we really only use one model. The simpler model in Figure 2C is a smaller network model. This model is amenable to mathematical analysis that serves to identify the role of the network motif and the relative contributions of AC and PDE (v(AC) and v(PDE) in Figure 2C) in governing the time delay between calcium and cAMP oscillations. Without incorporating all the reactions and the kinetic parameters, such a model can give us insight into the underlying behavior. What the reviewer identifies as the second model, is in fact, an expanded version of the first model, with detailed biochemical reactions and more parameters. In the revised manuscript, we clarified that there really is only one model that is presented in two different levels of detail.

Currently, the spatial model is fit by visual inspection to the experimental model for oscillation time, phase, and amplitude (assuming that fluorescence is proportional to concentration). These design decisions are now elaborated in Appendix 1. One of the challenges with using more stringent, quantitative fitting routines such as particle swarm or evolutionary programming, in this case, is the size and scale of the spatial resolution of the measurements. Despite our simple fitting approach, we are reasonably confident that the model captures the key physics in the spatial control of phase and oscillations of calcium-cAMP as demonstrated by our model validation and prediction.

The following two predictions where both models would yield diametrically opposing outcomes:i) The 3D reaction diffusion model predicts that cAMP oscillations in the AKAP79/150 compartments and cAMP oscillations in the bulk cytosol are out of phase in the same cell (see Figure 4). This can be tested by combining the AKAP79-based FRET sensor with a cytosolic, intensiometric, RFP-based cAMP sensor such as R-FlincA (Ohta et al., 2018). If the authors observe a clear phase shift between these two reporters, it would constitute very strong evidence in support of the 3D reaction-diffusion model. The current datasets only report one type of compartment per cell, yet the striking difference between compartments in the same cell is where the model predictions differ most dramatically.

We thank the reviewers for this suggestion to use the red R-FlincA and the cyan-yellow FRET-based Epac2-camps for multiplexed detection of cAMP dynamics. In previous attempts, we found it difficult to measure cAMP dynamics in more than one compartment in the same cell presumably due to binding competition effects. Despite the technical challenges, in this case, we observed anti-correlated cAMP oscillations in MIN6 cells expressing appropriate levels of cytosolic R-FlincA and AKAP79-localized Epac2-camps, suggesting the coexistence of the compartmentalized phase relationship, consistent with our model (Figure 1—figure supplement 2, n = 25 cells). We have added these data as Figure 1—figure supplement 2 in the revised manuscript.

ii) Localized oscillations depend on the assumption that AC diffusion is (close to) zero. Thus, the authors should measure AC diffusion at the plasma membrane. A key assumption of their model is that at least a fraction of the total protein pool should be completely immobile. Furthermore, this fraction should increase in an AKAP150 overexpression background.

We agree with the reviewer and have measured the AC8 diffusion at the plasma membrane. The measured diffusion coefficient value of 0.019 ± 0.002 μm^2^/sec is very low and indeed there is a significant immobile fraction of AC8 at the membrane (Figure 3—figure supplement 2), validating the model assumption. This data has been included as Figure 3—figure supplement 2 in the revised manuscript.

E) Final punchline that these elegant biophysical findings are important for some aspect of physiology is not very compelling. The authors should come with a more surgically-dissected physiological parameter that depends on the phase relationship between Ca and CaM, and at least provide this in their Discussion section.

We thank the reviewer for bringing this issue to our attention. We have now added discussions about the impact of the spatially-segregated oscillation phase relationship between Ca^2+^ and cAMP on β cell insulin secretion in the revised manuscript.